# Description and validation of the ice sheet model Nix v1.0

Daniel Moreno-Parada[1,2], Alexander Robinson[3,1], Marisa Montoya[1,2], and Jorge Alvarez-Solas[1,2]

[1]Departamento de Física de la Tierra y Astrofísica, Universidad Complutense de Madrid, Facultad de Ciencias Físicas, 28040 Madrid, Spain
[2]Instituto de Geociencias, Consejo Superior de Investigaciones Cientifícas-Universidad Complutense de Madrid, 28040 Madrid, Spain
[3]Alfred Wegener Institute, Helmholtz Centre for Polar and Marine Research, Potsdam, Germany

**Correspondence:** Daniel Moreno-Parada (danielm@ucm.es)

**Abstract.** We present a physical description of the ice-sheet model Nix v1.0, an open-source project intended for collaborative development. Nix is a two-dimensional (flowline plus a vertical dimension) thermomechanical model written in C/C++ that simultaneously solves for the momentum balance equations, mass conservation and temperature evolution. Nix's velocity solver includes a hierarchy of Stokes approximations: Blatter-Pattyn, depth-integrated higher order and shallow-shelf. The grounding-line position is explicitly solved by a moving coordinate system that avoids further interpolations. The model can be easily forced with any external boundary conditions. Nix has been verified for standard test problems. Here we show results for a number of benchmark tests from the Marine Ice Sheet Intercomparison Project (MISMIP) and assess grounding-line migration with an overdeepened bed geometry. Lastly, we further exploit the thermomechanical coupling by designing a suite of experiments where the forcing is a physical variable, unlike previously idealised forcing scenarios where ice temperatures are implicitly fixed via an ice rate factor. Namely, we use atmospheric temperatures and oceanic temperature anomalies to assess model hysteresis behaviour with active thermodynamics. Our results show that hysteresis in an overdeepened bed geometry is similar for atmospheric and oceanic forcings. We find that not only the particular sub-shelf melting parametrisation determines the temperature anomaly at which the ice sheet retreats, but also the particular value of calibrated heat exchange velocities. Notably, the classical hysteresis loop is widened for both forcing scenarios (i.e., atmospheric and oceanic) if the ice sheet is thermomechanically active as a result of the internal feedback among ice temperature, stress balance and viscosity. These results show that a temperature-dependent ice viscosity provides inertia and stability to the ice sheet, regardless of the particular external forcing applied. In summary, Nix combines rapid computational capabilities with a Blatter-Pattyn stress balance fully coupled to a thermomechanical solver, not only validating against established benchmarks but also offering a powerful tool for advancing our insight into ice dynamics and grounding-line stability.

# 1 Introduction

Marine ice sheets, such as the present-day West Antarctic Ice Sheet (WAIS), are of particular interest for the glaciological community and have been fundamental objects of study in the last decades (Payne et al., 2000; Pattyn et al., 2008; Bamber et al., 2009; Feldmann and Levermann, 2015; Shepherd et al., 2018; Martin et al., 2019; Rignot et al., 2019; Robel et al., 2019; Pattyn and Morlighem, 2020; Garbe et al., 2020; DeConto et al., 2021; Joughin et al., 2021; Hill et al., 2023). Since their bedrock lies mostly below sea level, they are prone to rapid changes (Bentley, 1998), leading a number of authors to question their stability (e.g., Bamber et al., 2009; Mouginot et al., 2014; Paolo et al., 2015; Feldmann and Levermann, 2015; Shepherd et al., 2018; Rignot et al., 2019; Robel et al., 2019; Pattyn and Morlighem, 2020; Garbe et al., 2020; Joughin et al., 2021; Hill et al., 2023). A complete WAIS collapse would imply 3-5 metres of level rise (Bentley, 1998; Fretwell et al., 2013), leaving the future of the WAIS as a key uncertainty for sea-level projections.

An accurate numerical description of the grounding line is thus fundamental for the reliability of such projections. A number of attempts have been made in the past to simulate grounding-line migration within marine ice-sheet models. Weertman (1974) and Thomas and Bentley (1978) proposed that no stable steady states of the grounding line could be found on inland-sloping or retrograde beds. Hindmarsh (1993a) later introduced the possibility of neutral equilibrium under the premise that the equilibrium position is continuous and hence there exists an infinite number of equilibrium configurations. More recently, Vieli and Payne (2005) assessed the influence of numerical details and discretization on the dynamics of the grounding line, concluding that a reliable method of treating grounding-line migration within numerical ice-sheet models was unknown. Later studies confirmed the possibility of numerical artefacts (Pattyn et al., 2006; Hindmarsh, 2006; Schoof, 2006a, b, 2011), in agreement with the early works of Weertman (1974) and Thomas and Bentley (1978). Even so, Vieli and Payne (2005) and Pattyn et al. (2006) hypothesised the possibility of "neutral equilibrium", first introduced by Hindmarsh (1993a). The analytical approach of Schoof (2007b) based on assymptotic expansions eventually concluded that these results were numerical artefact appearing for certain parameter regimes. Only in the absence of basal sliding has the possibility of non-unique steady states been raised (Nowicki and Wingham, 2008).

Amidst the lack of a reliable model of grounding-line migration, the first Marine Ice Sheet Intercomparison Project (MISMIP, Pattyn et al., 2012) shed light on the agreement of modelling efforts to describe the grounding-line motion and assessed the appropriateness of numerical schemes. The authors proposed a set of benchmark experiments on an idealised two dimensional bed geometry, concluding that moving grid models are the most reliable choice from a numerical perspective as the grounding line is part of the solution and no interpolations are required.

MacAyeal and Barcilon (1988) notably showed that a two-dimensional free-floating shelf has no effect on the dynamics of the grounded ice upstream of it (later underlined by Schoof, 2007b). As a result, a boundary condition can be directly imposed at the grounding line that is solely dependent on the ice thickness therein, irrespective of the particular shape or the dynamics of the shelf. A correct description of a 2D marine ice sheet thus relies on an appropriate formulation of the grounded ice dynamics, specially near the terminus position where ice streaming is generally found.

For a comparison with the semi-analytical solutions of Schoof (2007b), ice streaming (i.e., fast flowing ice due to basal sliding) becomes a necessary condition given that the boundary layer theory assumes rapid sliding near the grounding line. Ice streams are in fact a distinct feature of ice sheets with no counterpart in other geophysical thin-film flows. These regions of rapidly flowing ice exhibit velocities even three orders of magnitude faster than the usual glacial ice, yet they only account for a small fraction of the total ice-sheet area (e.g., less than $5\%$ of the Antarctic ice sheet; Bamber et al., 2000). Even so, it is important to represent them correctly to evaluate ice outflow discharge, ice-sheet sensitivity and overall stability.

The rapid flow of ice streams fails to be explained by vertical shearing of ice. In other words, friction at the bed is typically smaller than the driving stress predicted by a lubrication approximation (Whillans and van der Veen, 1997; Joughin et al., 2004). Rather, high ice-stream velocities are caused by the deformation of meltwater-saturated, weak subglacial till (Alley et al., 1986; Blankenship et al., 1986; Engelhardt et al., 1990), thus consistent with geophysical studies showing that basal sliding is fundamentally a sort of Coulomb slip connected with the mechanical failure of plastic till (e.g., Tulaczyk, 1999).

Schoof (2006b) later extended the work to depth-integrated viscous flows used in three-dimensional ice-sheet models. Namely, a variational formulation of the two-dimensional Shallow Shelf Approximation (SSA) equations is given without assumptions on the extension of the sliding domain. In fact, as noted by the author, sliding regions must be determined as part of the solution and are consequently not known *a priori*. Notably, a solvability condition was also derived (as in Schoof, 2006b) to guarantee the existence of physical solutions. Strictly speaking, if the till is too weak so that the total momentum of applied forces is greater than the maximum momentum of frictional force about a given point, then no solutions are expected to exist.

A variational formulation entails strong consequences both from a physical and a mathematical point of view. Particularly, it eludes explicit manipulation of the unknown sliding domain extension, additionally provides a numerical method for solving the ice flow problem and it ensures well-posedness of the SSA non-linear elliptic equations since they can be derived from a convex and bounded below functional (Schoof, 2006b). However, the time-evolving system of the SSA stress balance coupled to the advection equation is not yet known to be mathematically well posed (Bueler and Brown, 2009).

More recently, Goldberg (2011) derived a higher-order stress approximation using variational methods with similar accuracy to the Blatter-Pattyn momentum equations (Blatter, 1995; Pattyn, 2003), though differences are particularly notable for resolutions below 20 km. The velocity solver was first adapted for multimillenial 3D ice-sheet models CISM (Lipscomb et al., 2019), where this depth-integrated velocity approximation was referred to as DIVA. Nevertheless, the DIVA solver had been previously used in continental scale models by Arthern et al. (2015) and Arthern and Williams (2017). The numerical stability of this solver was systematically studied by Robinson et al. (2022), who show that the DIVA solver outperformed the remaining solvers in terms of both model performance and the representation of the ice-flow physics itself.

The appropriate stress balance treatment is merely one of the challenges of ice streaming and grounding-line stability. Understanding the mechanisms governing its temporal variability also remains as a major obstacle, particularly at the aim of developing models of ice-sheet dynamics (Robel et al., 2013). Given the broad range of ice-flow speeds observed in real ice sheets (e.g., Shepherd and Wingham, 2007; Truffer and Fahnestock, 2007; Vaughan and Arthern, 2007), numerical simulations of these rapidly flowing bands are a well-known difficulty, partially due to the fact that fast grounded ice flow is a combination of sliding over a hard/soft bed and shear deformation of the basal. Moreover, ice high-quality spatially distributed observations

of near-base conditions are rare and constraining models in fast flowing regions becomes challenging (Bueler and Brown, 2009). Various modelling approaches have been considered to correctly describe the large complexity of ice-stream dynamics. Tulaczyk et al. (2000a) found that subglacial hydrology yields multiple modes of ice stream flow in a highly reduced model. Parameterizations of observed small-scale phenomena (e.g., drainage networks) were later considered by coupling a flow-band model and a simple hydrological model (Bougamont et al., 2003; Bougamont and Tulaczyk, 2003). Another flow-band model was employed by van der Wel et al. (2013), additionally introducing a dynamic drainage model.

Two-dimensional models have tremendously helped to understand ice-sheet dynamics both from a theoretical (e.g., Weertman, 1974; Hindmarsh, 1993b; Chugunov and Wilchinsky, 1996; Hindmarsh, 1996; Schoof, 2005, 2006b, 2007b, a, 2011) and a modelling perspective. Numerous authors have contributed to the latter, thus demonstrating the practical use of a two-dimensional setup. Hindmarsh and le Meur (2001) assessed the dynamical processes involved in the retreat of marine ice sheets, with a particular interest in the WAIS the Last Glacial Maximum. Haseloff and Sergienko (2018) later considered the effect of buttressing on grounding-line dynamics, thus corroborating the findings of existing numerical studies that the stability of confined marine ice sheets is influenced by ice-shelf properties. Other 2D ice-sheet models additionally employ real bedrock geometry sections. This is the case of Pattyn et al. (2006), who studied the role of transition zones in marine ice-sheet dynamics, and Jamieson et al. (2012), where ice-stream stability was investigated on a reverse bed slope. The realism of the 2D setup can also account for glacial isostatic adjustment. To illustrate this, Payne (1995) studied limit cycles in the basal thermal regime of ice sheets considering a constant diffusivity of the asthenosphere. More recently, Bassis et al. (2017) investigated how Heinrich events are triggered by ocean forcing and modulated by isostatic adjustment, though the viscosity dependency on temperature was not considered. Other examples of simplified physics that neglect thermochemical coupling focused instead on attribution exercises of anthropogenic-induced ice-sheet retreat (e.g., Christian et al., 2022), consequently biased by unrealistic constant temperatures both in space and time. Lastly, even though ice shelves are not explicitly resolved in 2D models, the potential role of buttressing can also be considered via a parametrisation (e.g., Dupont and Alley, 2005; Schoof, 2007b; Jamieson et al., 2012; Robel et al., 2014, 2019).

Despite the extensive research on the topic, important questions regarding the particular effect of thermodynamics remain unanswered. Specifically, it is unclear whether marine ice sheets have discrete steady surface profiles if ice temperatures can freely evolve in time and what the potential potential implications would be for the hysteresis behaviour in overdeepened bed geometries. In ice streaming regions, ice flow occurs mostly along one main direction, thus becoming the preferred axis across which lateral variations are negligible. It is a common approach to reduce the number of horizontal dimensions to the main flow direction so as to minimize computing time while allowing for realistic applications. Nonetheless, the thermal state of the ice and the potential oceanic forcing are still fundamental pieces to understand the future evolution of ice sheets that have not been considered in low-dimensional models.

In this line, we herein introduce the 2D ice-sheet model Nix. Unlike previous 2D models, the default setup consists of a Blatter-Pattyn stress balance fully coupled to a thermodynamical solver that accounts for both vertical and along-flow horizontal heat advection, as well as vertical diffusion. Note that other configurations are also possible given the independent structure of Nix funtionalities. In other words, the user can select a particular stress balance description from a hierarchy of

Stokes approximations (i.e., Blatter-Pattyn, Depth Integrated Viscosity Approximation, SSA or SIA) and optionally solve the associated heat problem. The paper is structured as follows: we begin by describing the technical model design (Section 2). We then describe the physical approximations (Section 3) and numerics (Section 4) of the model. Several benchmarks and idealized experiments are presented in Section 5. A thorough discussion is given in Section 6 and the work is summarised in Section 7.

## 2 Model design

Nix is an open source software available under the Creative Commons Attribution 4.0 International license. The model has been derived from scratch with a clear Application Programming Interface (API). It is written in C/C++ for efficiency and extremely fast computing (see Appendix H) and is readily available to run in any High Performance Computing Cluster. There are two key dependencies: NetCDF (Rew and Davis, 1990; Brown et al., 1993) and Eigen (Guennebauda et al., 2010) libraries. The former handles tasks for convenient community-standard input/output capability, whereas the latter serves to define vectors, matrices and further necessary computations (Fig. 1). Nix users can optionally select parallel computing (supported by Eigen library) simply by enabling OpenMP on the employed compiler, particularly convenient for high resolutions in the Blatter-Pattyn approximation, where large sparse matrices must be inverted. Moreover, it is also possible to use Eigen's matrices, vectors, and arrays for fixed size within CUDA kernels (Nickolls et al., 2008).

Nix's design offers a friendly Python wrapper module that handles directory management and compilation, though it can be compiled and run independently. The exact version used to produce the results of this work is archived at a persistent Zenodo repository (Moreno-Parada et al., 2023) while the latest version can be accessed on GitHub at: https://github.com/d-morenop/nix.

As a result, Nix presents itself as an extremely versatile model combining usage simplicity, low computational costs and high-order physics at extremely high resolution ($\Delta x < 0.1$ km). Moreover, the practical use of Nix ice-sheet model does not solely lie on its high-resolution performance, but also in the gap filled in the model hierarchy spectrum: a 2D model solving for the higher-order Blatter-Pattyn stress balance, thermodynamically coupled and computationally inexpensive.

## 3 Model physics

In this section, the fundamental equations of the model are described. Generally speaking, we consider an ice slab of two spatial dimensions (i.e., horizontal and vertical) by coupling a particular choice of stress balance, the advection equation and the associated heat problem.

Our system evolves thermodynamically in time through three main processes concerning heat propagation: vertical diffusion, horizontal and vertical advection and internal deformation of the ice. Viscosity is thus dependent on both the strain rate and the temperature. With respect to dynamics, basal friction can be parametrised by three distinct formulations (linear, power-law and

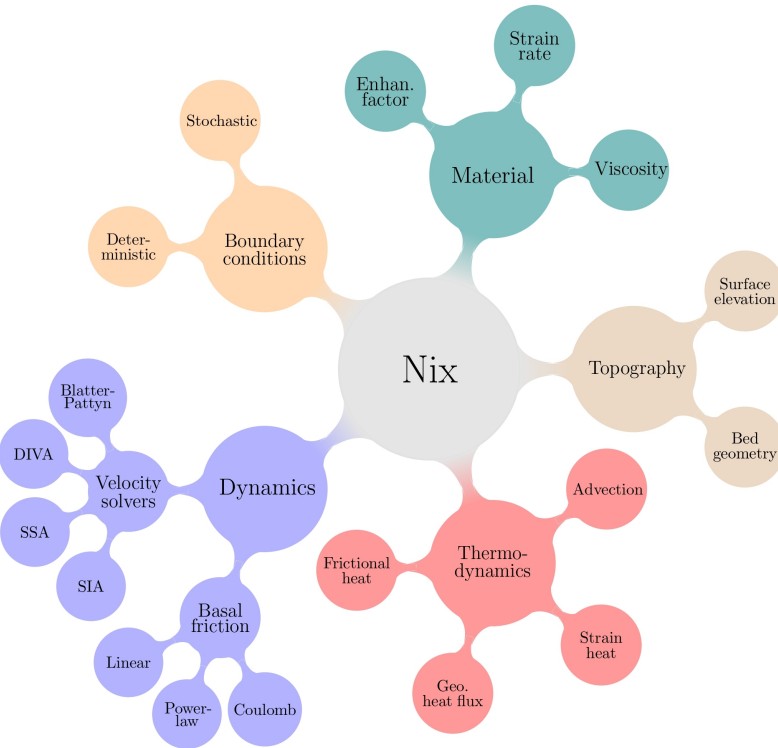

**Figure 1.** Overview of Nix modular structure. Each colour represents a C++ class: dynamics, material, topography, thermodynamics and boundary conditions. The Python wrapper is an optional user friendly option and the code can be compiled without any additional dependencies at any standard High Performance Computing Cluster.

Regularized-Coulomb). Additionally, basal friction captures the thermal state of the base by a two-valued friction coefficient encapsulating frozen and thawed bedrocks.

### 3.1 The Blatter-Pattyn approximation

Ice sheets and glaciers are generally described as an incompressible fluid with a low Reynolds number flow. Conservation of momentum is ensured through the Stokes equations, a quasi-static stress description where intertial and advective terms are neglected due to the slow movement of the ice.

The typical ice-sheet geometry allows to further simplify the Stokes flow equations by defining an aspect ratio $\varepsilon$. Given the characteristic length scales for the horizontal and vertical dimensions, $\varepsilon \ll 1$ (e.g., Greve and Blatter, 2009). Simply by keeping terms of order $\mathcal{O}(\varepsilon)$ in the Stokes equations, the Blatter-Pattyn model (Blatter, 1995; Pattyn, 2003) arises with a hydrostatic approximation error of $\mathcal{O}(\varepsilon^2)$ (Dukowicz et al., 2010; Schoof and Hindmarsh, 2010). This first-order approximation forms an elliptic coercive problem, significantly easier to solve than the intricate saddle-point problem of the full Stokes system.

For the purpose of this work, we shall consider two spatial dimensions: horizontal $x$ and vertical $z$, respectively. This considerably reduces the computational time and allows for extremely high spatial resolutions ($\Delta x \sim 0.5$ km), whilst explicitly accounting for the vertical gradients in ice viscosity and velocity. The 2D version of the Blatter-Pattyn model can be written as:

$$\frac{\partial}{\partial x}\left(4\eta\frac{\partial u}{\partial x}\right) + \frac{\partial}{\partial z}\left(\eta\frac{\partial u}{\partial z}\right) = \rho g \frac{\partial h}{\partial x}, \tag{1}$$

where $\rho$ is the ice density, $g$ is the gravitational acceleration, $\eta(x,z)$ is the effective viscosity, $h(x)$ is the surface elevation and $u(x,z)$ is the ice velocity.

The problem is subjected to a set of boundary conditions. Nix considers potential friction at the base of the ice, and a free surface on the upper boundary (Veen and Whillans, 1989). In terms of velocity gradients, the free surface condition can be expressed as (Pattyn, 2003):

$$\frac{\partial u}{\partial z} = 4\frac{\partial u}{\partial x}\frac{\partial h}{\partial x}, \tag{2}$$

and the basal drag is defined as the sum of all resistive forces:

$$\frac{\partial u}{\partial z} = 4\frac{\partial u}{\partial x}\frac{\partial b}{\partial x} + \frac{\tau(u)}{2\eta}, \tag{3}$$

for the base ($z = b$) in the presence of potential drag $\tau(u)$ (see Section 3.4 for a thorough description on basal friction). A stress-free base can be obtained simply by setting $\tau = 0$ in Eq. 3.

We further assume an ice divide at one end of the domain ($x = 0$), where $\partial u/\partial x = 0$, and hydrostatic equilibrium at the shelf-ocean boundary ($x = L$), where the water pressure balances the longitudinal stress gradient. The full problem thus takes the following succint form (subscripts hereinafter denotes partial differentiation):

$$\begin{cases} (4\eta u_x)_x + (\eta u_z)_z = \rho g h_x, & x \in \mathcal{I},\ z \in \mathcal{L}, \\ u_z = 4u_x h_x, & x \in \mathcal{I},\ z \in \partial\mathcal{L}^+, \\ 2\eta u_z = 8\eta u_x b_x + \tau, & x \in \mathcal{I},\ z \in \partial\mathcal{L}^-, \\ u_x = 0, & x = 0,\ z \in \mathcal{L}, \\ 8\eta u_x = \rho g H^2 - \rho_w g z^2, & x = L,\ z \in \mathcal{L}, \end{cases} \tag{4}$$

where $\rho_w$ is the water density, $H$ is the ice thickness evaluated at the grounding line $x = L$ and the $\partial\mathcal{L}^\pm$ symbols denote the upper and lower vertical boundaries, respectively.

## 3.2 The Depth Integrated Viscosity Approximation

We now briefly describe the mathematical problem underlying the Depth Integrated Viscosity Approximation (DIVA) stress balance in Cartesian coordinates (Goldberg, 2011; Lipscomb et al., 2019).

As for the Blatter-Pattyn model, we shall consider only one horizontal dimension, leaving unaltered the nature of DIVA/SSA equations. This allows us to consider our model as a longitudinal section of a three-dimensional ice stream:

$$\frac{\partial}{\partial x}\left(4\bar{\eta}H\frac{\partial u}{\partial x}\right)+\tau(u)=\rho gH\frac{\partial h}{\partial x}. \tag{5}$$

Since the stress balance is also a second-order partial differential equation on the velocity, we again need two boundary conditions. Analogously to the Blatter-Pattyn approximation, we assume an ice divide at one end. At the other end of the domain, the problem is subjected to a dynamic boundary condition that accounts for the balance between cryostatic and hydrostatic pressures. Thus, we can express the DIVA/SSA boundary problem in the following compact form:

$$\begin{cases} (4\bar{\eta}Hu_x)_x + \tau(u) = \rho gHh_x, & x \in \mathcal{I}, \\ u = 0, & x = 0, \\ 8\bar{\eta}u_x = \rho gH^2 - \rho_w gD^2, & x = L, \end{cases} \tag{6}$$

where $D$ is the distance from the sea surface to the bottom of the ice.

Equation 5 is an elliptic non-linear differential equation. In the purely SSA form (neither velocity nor viscosity dependency on $z$, i.e., $u = \bar{u}$ and $\eta = \bar{\eta}$), it constitutes the simplest form of longitudinal stress balance derivable from the Stokes model (Bueler and Brown, 2009). We can then solve for the velocity $u(x)$ by integrating Eq. 6 if the functions $H(x)$, $\bar{\eta}(x)$, $h(x)$ and $\tau(u)$ are known. We have implemented an implicit algorithm so as to numerically integrate the one-dimensional DIVA/SSA equation (see integrating scheme description in Appendix C).

### 3.3 The advection coupling

Given that all models herein presented provide a quasi-static description of the ice flow, the stress balance does not determine the temporal evolution of the system, but rather it represents an equilibrium state for a particular ice thickness $H(x)$ and viscosity $\eta(x,z)$ configuration. The temporal evolution is generally considered by coupling the stress balance to the advection equation:

$$H_t + (\bar{u}H)_x = S(x), \tag{7}$$

where $S(x)$ is the surface mass balance. Given that Eq. 7 is first order, we only need one boundary condition $H(x=0,t)$ and the consequent initial condition $H(x,t=0)$.

We now couple Eqs. 6 and 7 to study the evolution of the ice thickness $H(x,t)$ governed by the advection equation, where the velocity field $u(x,z)$ satisfies the stress balance imposed by a particular choice of the Stokes approximation. Namely, our

problem takes the following mathematical form:

$$\begin{cases} H_t + (\bar{u}H)_x = S(x), & x \in \mathcal{I},\ t > 0 \\ (4\eta u_x)_x + (\eta u_z)_z = \rho g h_x, & x \in \mathcal{I},\ z \in \mathcal{L}, \\ H = H_0, & x \in \mathcal{I},\ t = 0. \\ h_x = 0, & x = 0,\ t > 0. \\ u = 0, & x = 0,\ t > 0. \\ 8\eta u_x = \rho g H^2 - \rho_w g z^2, & x = L,\ z \in \mathcal{L},\ t > 0. \end{cases} \tag{8}$$

From a purely physical perspective, Eq. 8 describes a fluid membrane of variable thickness driven by its own weight that evolves in time due to advection.

## 3.4   Basal friction

Basal shear stress can be generally expressed as a function of the sliding velocity $u_b$ and the effective pressure $N$, i.e., $\tau_b = f(u_b, N)$. The physical properties of the material over which the ice may potentially slide can correspond either to a hard

bedrock flow (e.g., Weertman, 1957) or to a Coulomb-plastic rheology (e.g., Tulaczyk et al., 1998). Moreover, the influence of the sliding velocity on $\tau_b$ is often represented by a power friction law, although a regularization term $u_0$ accounting for local properties of the bed has been shown to outperform such a power law in both pressurized ice experiments (Zoet and Iverson, 2020) and observations (Minchew et al., 2018; Stearns and van der Veen, 2018; Joughin et al., 2019)

    As a result, Nix can calculate the basal shear stress (i.e., basal drag) via two independent formulations: a pseudo-plastic

power law (Schoof, 2010; Aschwanden et al., 2013) and the regularized-Coulomb law (Schoof, 2005; Joughin et al., 2019). The former reads:

$$\boldsymbol{\tau}_b = -c_b \left( \frac{|\boldsymbol{u}_b|}{u_0} \right)^q \frac{\boldsymbol{u}_b}{|\boldsymbol{u}_b|}, \tag{9}$$

where $u_0 = 100$ m/yr and $c_b$ is a spatially-variable friction coefficient defined below. We shall focus on two particular cases of the pseudo-plastic law based upon the choice of the exponent $q$. Namely, the linear law ($q = 1$; e.g., Quiquet et al., 2018) and

the purely plastic law ($q = 0$).

    On the other hand, the regularized-Coulomb formula is given by:

$$\boldsymbol{\tau}_b = -c_b \left( \frac{|\boldsymbol{u}_b|}{|\boldsymbol{u}_b| + u_0} \right)^q \frac{\boldsymbol{u}_b}{|\boldsymbol{u}_b|}, \tag{10}$$

behaving as a power law for small sliding velocities (i.e., $u_b < u_0$) whilst always yielding a bounded friction value for arbitrarily high velocities (i.e., $u_b \gg u_0$). Following Zoet and Iverson (2020), we set $q = 1/5$ and $u_0 = 100$ m/yr by default to ensure a

reasonable transition to the steady-state shear stress supported by the till bed. The same study empirically established that $q$ remains unaffected by variations in the detailed bed surface geometry.

The basal drag coefficient $\beta$ is usually defined as:

$$\beta = c_b(x)N, \tag{11}$$

where $N = \rho g H$ is the overburden pressure exerted by the ice column and $c_b(x)$ is a coefficient that reflects the bedrock characteristics.

Nevertheless, for simplicity and consistency with prior benchmark experiments as MISMIP (Pattyn et al., 2012), the model also allows to represent basal friction as:

$$\tau_b = Cu^q, \tag{12}$$

with the chosen value of $C = 7.624 \times 10^6$ Pa m$^{-1/3}$ s$^{1/3}$ and $q = 1/3$, a sliding velocity of about 35 m yr$^{-1}$ yields a basal shear stress of 80 kPa.

## 3.5 Thermodynamics

The ice temperature in the flow line depends on the two spatial dimensions $x$ and $z$ (horizontal and vertical, respectively) along with time (i.e., $\theta = \theta(x, z, t)$). Heat transfer is further considered to occur due to vertical diffusion, both horizontal and vertical advection and internal heat deformation. Energy conservation is ensured in a classical approach by a balance equation that neglects neglecting horizontal diffusion (Greve and Blatter, 2009):

$$\begin{cases} \rho c \theta_t = k\theta_{zz} - \rho c(u\theta_x + w\theta_z) + \Phi, & x \in \mathcal{I},\ z \in \mathcal{L},\ t > 0, \\ \theta = \theta_0, & x \in \mathcal{I},\ z \in \mathcal{L},\ t = 0, \\ \theta_z = -G/k, & x \in \mathcal{I},\ z = \partial\mathcal{L}^-,\ t > 0, \\ \theta = \theta_L, & x \in \mathcal{I},\ z = \partial\mathcal{L}^+,\ t > 0, \end{cases} \tag{13}$$

where $k$ is the ice conductivity, $c$ is the specific heat capacity, $\Phi = 4\eta\dot{\varepsilon}^2$ denotes the internal strain heating, $G$ is the geothermal heat flow, $\theta_0$ is the initial temperature profile and $\theta_L$ surface ice temperature. The $\partial\mathcal{L}^{\pm}$ symbols denote the upper and lower vertical boundaries, respectively.

The energy balance is discretised using an upwind scheme with a forward Euler step and centred differences for the spatial derivatives (see Appendix A4 for a detailed description).

## 3.6 Viscosity

We consider Glen's flow law (Glen 1955; Nye, 1957) to relate the shear stress, the ice temperature and the pressure of isotropic polycrystaline ice. Formation of anistropic fabric is considered via a flow enhancement factor.

As shown in Section 3.1, the Blatter-Pattyn stress balance equations define the effective viscosity as:

$$\eta = \frac{B}{2}\left(\dot{\varepsilon}^2 + \dot{\varepsilon}_0^2\right)^{\frac{1-n}{2n}}, \tag{14}$$

where $B$ is the ice hardness, $n = 3$ is the exponent in Glen's flow law, $\dot{\varepsilon}^2$ is the effective strain rate and $\dot{\varepsilon}_0^2$ is a regularization factor to elude potential singularities when velocity gradients are zero. Notably, for a 2D model with explicit thermodynamics, the viscosity expression further simplifies the expression of $B$ and $\dot{\varepsilon}^2$:

$$B = A(\theta)^{-1/n}, \tag{15}$$

$$\dot{\varepsilon}^2 = \left(\frac{\partial u}{\partial x}\right)^2 + \frac{1}{4}\left(\frac{\partial u}{\partial z}\right)^2, \tag{16}$$

where $n = 3$ is the Glen-flow exponent. $A(\theta)$ is the rate factor and follows an Arrhenius law:

$$A(\theta) = A_0 E e^{-Q/R\theta}, \tag{17}$$

$A_0$ and $Q$ are the temperature-dependent rate factor coefficient and activation energy, respectively (Greve and Blatter, 2009). $E_f$ is the so-called enhancement factor, commonly used to approximate the effect of anisotropic flow. It is possible to specify different values of the enhancement factor for different flow regimes (shear or stream). Typical values of the enhancement factor for the shearing and streaming regimes are $E_{\mathrm{shr}} = 3.0$ and $E_{\mathrm{strm}} = 0.7$ (Ma et al., 2010), respectively. Here we use for both a default value of $E = 1.0$.

For the vertically-integrated stress balance models (i.e., DIVA and SSA), Eqs. 14 and 15 are slightly modified by computing the vertically averaged quantities $\bar{\eta}$ and $\bar{B}$ following the generic formula $\bar{f} = \frac{1}{H}\int_b^h f dz$.

## 3.7 Grounding line

Nix aims at simulating the flow of a sliding ice sheet. Since the longitudinal stress at the grounding line $x = L$ is simply a function of the ice thickness therein $H(x = L)$ for a 2D ice sheet (Schoof, 2007b), the behaviour of grounded ice and the location of the grounding-line itself are completely independent of the floating part.

Neither the potential distinct shapes of the ice shelf (e.g., due to sub-shelf melting) nor the calving affect the dynamics of grounded ice. Thus, the flotation condition and the stress condition (Eq. 8) can be considered as boundary conditions at the grounding line. These two conditions are in fact sufficient to study the ice thickness evolution and the grounding-line migration.

Following Hindmarsh (1996), an explicit expression for the grounding-line migration rate $\dot{L}$ can be readily obtained from a total differentiation of the flotation condition:

$$\dot{L} \equiv \frac{dL}{dt} = \frac{\varrho D_t + (\bar{u}H)_x - S}{H_x - \varrho D_x}, \tag{18}$$

where $D$ is the water depth at the grounding line and $\varrho = \rho_w/\rho$ is the water-to-ice density ratio, respectively.

More recent studies suggest that the maximum terminus thickness is bounded by the yield strength of ice $\tau_c$ (Bassis and Walker, 2012; Bassis and Jacobs, 2013). Hence, a maximum ice thickness at the terminus occurs when the stress exceeds the depth integrated strength of ice:

$$H^{\mathrm{max}} = \frac{\tau_c}{\rho g} + \sqrt{\left(\frac{\tau_c}{\rho g}\right)^2 + \varrho D^2}, \tag{19}$$

thus constraining the terminus thickness such that $H(x = L, t) \leq H^{\max}$.

This approach eludes semi-empirical parametrizations of the calving (as in Schoof, 2007) and further provides a lower bound on the rate of grounding line advance (Bassis et al, 2017). Combining the continuity equation and the material derivative of $H^{\max}$ (Eq. 19), an expression for the rate of advance/retreat of the terminus can be readily obtained:

$$\frac{dL}{dt} \geq \frac{H_t}{H_x^{\max} - H_x},$$

(20)

at $x = L$. Negative sign indicates retreat.

Inequality Eq. 20 is analogous to the grounding-line migration derived for a marine ice sheet by Schoof (2007a; 2007b). Particularly, if $H^{\max}$ is given by the flotation condition, Eq. 20 exactly reproduces the grounding line position derived by Schoof (2007b) (Bassis et al., 2017).

## 3.8 Sub-shelf melting parametrization

Oceanic melting beneath ice shelves is the main driver of the current mass loss of the Antarctic ice sheet. For this reason, Nix considers various melting parameterisations, such as simple scaling with far-field thermal driving (e.g., Favier et al., 2019).

We adhere to local yet physically-based parametrizations based on ocean circulation models (Grosfeld et al., 1997). Namely, the linear dependency can be expressed as:

$$M = \gamma_T \varrho \frac{c_{\mathrm{po}}}{L_{\mathrm{i}}} (T - T_0),$$

(21)

where $\gamma_T$ is the heat exchange velocity, $T_0$ is a reference temperature, $c_{\mathrm{po}}$ is the specific heat capacity of the ocean mixed layer and $L_{\mathrm{i}}$ is the latent heat of fusion of ice.

This linear formulation with a constant exchange velocity $\gamma_T$ assumes a circulation in the ice-shelf cavity that is independent from the ocean temperature. This assumption is neither supported by modelling (Holland et al., 2008; Donat-Magnin et al., 2017) nor by observational studies (Jenkins et al., 2018) that suggest a larger circulation in response to a warmer ocean, subsequently increasing melt rates. One manner to account for this positive feedback is by considering a quadratic dependency (Holland et al., 2008):

$$M = \gamma_T \left( \varrho \frac{c_{\mathrm{po}}}{L_{\mathrm{i}}} \right)^2 (T - T_0)^2.$$

(22)

These two parametrizations have been employed in numerous studies (e.g., review in Asay-Davis et al., 2017; Favier et al., 2019). This melt rate is included as an additional frontal ablation term in the ice flux computation (Eq. D3) and it amounts to an additional outflow of ice beyond the grounding line velocity provided by the stress balance. By default, Nix uses a quadratic parametrization.

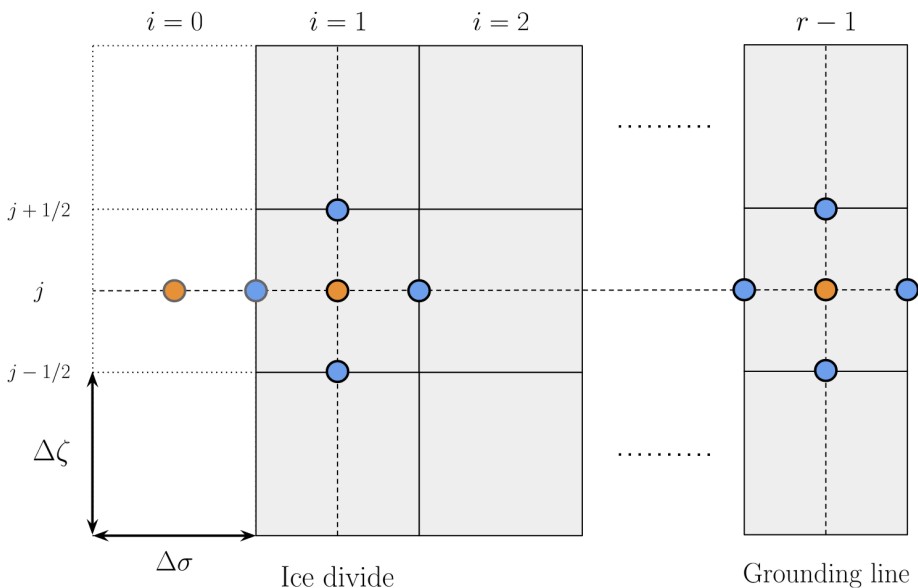

**Figure 2.** Nix staggered grid definition follows an Arakawa-C scheme (Arakawa and Lamb, 1977). The number of grid points the the horizontal $r$ and vertical $p$ is fixed in time (see Appendix A). As we employ a moving grid, the position of the last horizontal point $(r-1/2)$ explicitly tracks the grounding line $L(t)$. The grid spacing in the vertical $\Delta\zeta_j$ and horizontal $\Delta\sigma_i$ axis can be spatially dependent as Nix allows for nonuniform grids. Ghost points required to satisfy the boundary condition at the ice divide are noted in grey edge colour.

## 4 Model numerics

### 4.1 Moving grid transformation

Nix uses a nonuniform moving spatial grid that explicitly solves the grounding-line position. By default, the grid points distribution yields higher resolution near the grounding line following a polynomial or an exponential law (details in Appendix A). Evenly-spaced grids are also possible by setting the polynomial order to one.

As already noted by Pattyn et al. (2012), moving grid models are presumably the best choice in 2D models from a numerical perspective, as the grounding line position $L(t)$ is part of the solution and no interpolations are required. Given that neither the 325 terminus position $L(t)$ (i.e., the grounding line) nor the ice thickness $H(x,t)$ are fixed in time, we adopt a moving grid to trace their positions:

$$\sigma = \frac{x}{L(t)}, \quad \zeta = \frac{z - b(x)}{H(x,t)}, \quad \tau = t, \tag{23}$$

thus mapping the time-dependent intervals $0 \le x \le L(t)$ and $0 \le z \le H(x,t)$ into fixed ones $0 \le \sigma \le 1$ and $0 \le \zeta \le 1$. The variable $\tau$ is merely introduced to distinguish partial derivatives defined holding both $\sigma$ and $\zeta$ constant (as opposed to holding 330 $x$ and $z$ constant).

As a result, the corresponding derivatives contain additional terms (upon application of the Leibniz rule):

$$\frac{\partial}{\partial z} = \frac{1}{H}\frac{\partial}{\partial \zeta}, \tag{24}$$

$$\frac{\partial}{\partial x} = \frac{1}{L}\frac{\partial}{\partial \sigma} - \frac{1}{H}\left[(1-\zeta)\frac{\partial b}{\partial \sigma} + \zeta\frac{\partial H}{\partial \sigma}\right]\frac{\partial}{\partial \zeta}, \tag{25}$$

$$\frac{\partial}{\partial t} = \frac{\partial}{\partial \tau} - \frac{\sigma}{L}\frac{\partial L}{\partial \tau}\frac{\partial}{\partial \sigma} - \frac{\zeta}{H}\frac{\partial H}{\partial \tau}\frac{\partial}{\partial \zeta}. \tag{26}$$

For simplicity and analogously for the Blatter-Pattyn approximation, the advection equation coupled with the SSA/DIVA stress balance can be written in terms of the new variables. Thus, Eq. 8 reads:

$$\begin{cases} LH_\tau - \sigma\dot{L}H_\sigma + (uH)_\sigma = LS(\sigma,\tau), & \sigma \in \tilde{\mathcal{I}}, \ \tau > 0 \\ (4\bar{\eta}Hu_\sigma)_\sigma + \tau L^2 = L^2\rho g h_\sigma & \sigma \in \tilde{\mathcal{I}}. \\ H = H_0, & \sigma \in \tilde{\mathcal{I}}, \ \tau = 0. \\ H_\sigma = 0, & \sigma = 0, \ \tau > 0. \\ u = 0, & \sigma = 0, \\ 4\eta u_\sigma = \left(\rho g H^2 - \rho_w g D^2\right)L/2, & \sigma = 1, \end{cases} \tag{27}$$

where $\tilde{\mathcal{I}} \in [0,1]$ is the transformed interval and the subscripts denote partial differentiation.

Likewise, the third evolution equation that determines the behaviour of our system (i.e., the energy balance, Eq. 13) can be readily obtained in terms of our new variables:

$$\begin{cases} \rho c\left[L\theta_\tau - \sigma\dot{L}\theta_\sigma - \zeta LH_\tau\theta_\zeta/H\right] = \\ \quad kL\theta_{\zeta\zeta}/H^2 - \rho cu\left[\theta_\sigma - (b_\sigma + \zeta H_\sigma)\theta_\zeta/H\right] + L\Phi, & \sigma \in \tilde{\mathcal{I}}, \ \zeta \in \tilde{\mathcal{L}}, \ \tau > 0, \\ \theta = \theta_0, & \sigma \in \tilde{\mathcal{I}}, \ \zeta \in \tilde{\mathcal{L}}, \ \tau = 0, \\ \theta_z = -G/k, & \sigma \in \tilde{\mathcal{I}}, \ \zeta = \partial\tilde{\mathcal{L}}^-, \ \tau > 0, \\ \theta = \theta_L, & \sigma \in \tilde{\mathcal{I}}, \ \zeta = \partial\tilde{\mathcal{L}}^+, \ \tau > 0, \end{cases} \tag{28}$$

where the transformed intervals are again denoted by $\tilde{\mathcal{I}}$ and $\tilde{\mathcal{L}}$ respectively.

## 4.2 Spatial integration

### 4.2.1 Implicit scheme and Picard iteration

The lateral boundary condition is in fact non-trivial to implement using an explicit scheme (e.g., a shooting-like method) since it depends on the first spatial derivative of the velocity at the terminus position $\sigma = 1$, which might lead to convergence issues. Nix thus includes an alternative velocity solver based on an implicit discretization shcheme of all stress balance models described in Section 3 (numerical details in Appendix 3).

To account for the potential non-linearity in the velocity as a consequence of the viscosity and basal friction $\tau(u)$, the implicit solver uses a initial guess $\tau_0$ and $\eta_0$ and then enters a Picard iteration (see Theorem 2.2 in Teschl, 2012). A solution is hence obtained when the convergence criterion:

$$\frac{||u^n - u^{n-1}||}{||u^n||} < \phi_{\text{tol}} \tag{29}$$

is satisfied. The tolerance $\phi_{\text{tol}}$ can be set by the user but the default value is $10^{-6}$.

For the Blatter-Pattyn approximation, a sparse matrix must be solved in each Picard iteration. To do so, we apply a Bi-conjugate Gradient Stabilized method (commonly known as BiCGSTAB) with an Incomplete preconditioner (ILUT). On the contrary, the DIVA/SSA approximation solely requires solving a tridiagonal matrix at in each Picard iteration step, where the ice viscosity is updated. A tridiagonal solver algorithm is implemented as a subroutine within Nix to avoid additional external dependencies (see Appendix A).

### 4.3  Time integration

Once the velocity field $u(x,z)$ is obtained for a given set of boundary conditions and a particular ice thickness initial distribution $H(\sigma, \tau_0)$, we can compute the time evolution of the latter as a consequence of the advection imposed by $\bar{u}(x)$ and the surface mass balance $S(x,t)$ (Eq. 7). Thus, this coupled system formed by the momentum conservation and the continuity equation (Eq. 27) is fully integrated in two steps: first, a spatial integration to obtain the velocity velocity (where the ice viscosity is known); and then, a forward time integration to determine the new ice thickness. Lastly, the energy balance equation is integrated to compute the new temperature field.

Specifically, for a given initial ice-thickness distribution $H(x, t_0)$, the stress balance equation is spatially integrated, thus yielding the velocity $u(x, z)$. Then, the solution $u(x, z)$ (at $t_0$) and $H(x, t_0)$ allow us to integrate the continuity equation forward in time, consequently obtaining $H(x, t_0 + \Delta t)$. Additionally, this new ice thickness distribution yields $\theta(x, z, t_0 + \Delta t)$, thus constituting a self-consistent iterative method.

## 5  Methods and experimental set-up

Prior to any comprehensive description of the results, we must test whether Nix is capable of reproducing the benchmark tests of the Marine Ice Sheet Model Intercomparison Project (MISMIP Pattyn et al., 2012). To this end, we will perform all three MISMIP experiments: relaxation to steady state on a downward-sloping bed (Exp. 1), reversal of parameter (Exp. 2) and hysteresis on an overdeepening bed (Exp. 3). The aim of Exp. 1 was to show that there should be a single stable equilibrium profile on a downward-sloping bed. A backwards parameter relaxation in Exp. 2 was intended to demonstrate that grounding-line positions should be identical during advance and retreat, as steady states are unique. Exp. 3 was designed to assess whether ice-sheet models exhibit hysteresis behaviour and has become a benchmark for testing the capability of numerical models to simulate grounding-line migration.

First, we adopt the exact same problem definition so as to perform a one-to-one comparison. Next, we run an ensemble of simulations to address the question of whether the hysteresis with respect to model parameters variations found in in MISMIP Exp. 3 is still present even if the thermal state of the ice can evolve in time (as opposed to the idealised constant ice factor A set in Pattyn et al., 2012). Fixing $A$ uniquely determines a constant ice temperature, since $A(T)$ is a bijective function of the temperature. We therefore impose an atmospheric forcing (i.e., the ice surface boundary condition) that spans a wide range of realistic temperatures. As the geothermal heat flux provides a positive energy contribution, we expect a different thermal equilibrium profile for each imposed surface temperature. This yields a different viscosity field for each scenario, consequently leading to a different equilibrium velocity. As noted by Sergienko et al. (2013), the temperature profile is mostly determined by horizontal advection in streaming regions, thus bringing forward a strongly non-linear feedback worthy of attention.

Lastly, we force the system via ocean temperature anomalies with respect to a reference value $T_0$, so that $\Delta T_{\text{oce}} = T_{\text{oce}} - T_0$, whilst holding constant the air temperature throughout the simulation. We then convert these temperature anomalies into sub-shelf melting at the grounding line (e.g., Favier et al., 2019) by computing any of the parametrizations described in Section 3.8. Even though the air temperature is held constant (i.e., the boundary condition of our heat problem), the thermal state of the ice may evolve as both the thickness and extent are perturbed by the changing sub-shelf melting at the grounding line. Our particular ocean forcing consist of steps of $0.5$ºC evenly-spaced in time by 30 kyr to ensure equilibration, from $\Delta T_{\text{oce}} = 0$ to a maximum applied anomaly of 7ºC. Then, we reverse the forcing to recover the unperturbed state (i.e., zero anomaly).

It is worth noting that the basal friction remains identical to that in the MISMIP experiments both for the atmospheric and oceanic forcings. This means that no additional dependency of friction on temperature or hydrology is considered.

**Table 1.** Nix suite of experiments. The first row replicates MISMIP benchmark tests, whereas MISMIP-therm explores the hysteresis behaviour of a thermomechanically active ice sheet in two different forcing scenarios: amospheric and oceanic.

| Experiment name | Forcing variable | Thermodynamics | Melting/calving at GL |
| --- | --- | --- | --- |
| MISMIP (Exp. 1, 2 and 3) | Ice rate factor $A$ | No | No |
| MISMIP+therm (Exp. 3) | Air temperature $T_{\text{air}}$ | Yes | No |
|  | Ocean temperatures anom. $\Delta T_{\text{oce}}$ | No ($A = $ const.) | Yes |
|  |  | Yes ($A = f(T)$) | Yes |

# 6 Results

All simulations shown herein ran at a horizontal resolution of $\Delta x = 2$ km and 35 vertical layers. The particular mesh employed is unevenly-spaced with an increasing density of points towards the base and near the grounding line, following an exponential relation (see Appendix A).

## 6.1 MISMIP benchmark experiments

As a performance test for the Nix ice sheet model, simulations (Fig. 3) fairly reproduce the results shown by models that employ a stretched coordinate system as ours.

Figure 3b shows both the advancing and retreating phase in Pattyn et al. (2012) (Experiments 1 and 2, respectively), equilibrium grounding-line positions coincide and points thus overlap. Additionally, we find no stable equilibrium states for the downward-sloping bed of Exp. 3 (Fig. 3c and 3c), in agreement with the theoretical considerations by Weertman (1974) and Schoof (2007b). Namely, as we gradually decrease $A$ (i.e., increasing ice viscosity), the grounding-line position advances across the downward sloping bed until the upward-sloping region is reached (Fig. 3c). The ice flux at the grounding line then continuous to increase as $A$ decreases. Illustrated by Fig. 3d, when the ice flux is large enough so that there exists a stable solution beyond the unstable region (at the right-hand side of the bedrock peak in Fig. 3c), the grounding line traverses the upwards-sloping sector reaching a new stable solution.

Additionally, we repeat the three MISMIP experiments using the more sophisticated velocity solvers available in Nix: DIVA and Blatter-Pattyn. A direct inspection of Figs. 3b and 3d reveals that the solutions are nearly identical to the simpler SSA version, both for a downwards sloping and the overdeepending beds. The hysteresis is particularly well captured in all three Stokes approximations. Thus, we will use the SSA solver in the remainder of the current work to minimise computational costs unless otherwise stated.

## 6.2 MISMIP + thermodynamics

To exploit the fact that Nix is fully coupled with a thermodynamic solver, we further investigate the equilibrium states (Schoof, 2007b) when the system is forced via two different forcings: air temperatures $T_{\mathrm{air}}$ and ocean temperature anomalies $\Delta T_{\mathrm{oce}}$. Both describe more realistic conditions with slight variations. The former implies the same underlying perturbation mechanism (as for the idealised rate factor $A$ forcing): temperature changes within the ice modify its viscosity so that the grounding line migrates to reach a new equilibrium position. Nevertheless, when forcing the system with ocean temperature anomalies $\Delta T_{\mathrm{oce}}$ while keeping the air temperature constant, we perturb the system via an additional outflow term at the grounding line. By separately studying each mechanism, we can determine whether a marine terminating ice sheet might undergo hysteresis under different forcings.

### 6.2.1 Air temperature $T_{\mathrm{air}}$ forcing

At the aim of building a thermomechanically active version of MISMIP experiments, the natural choice is to convert the idealised ice-rate factor forcing in MISMIP into temperatures (via an Arrhenius law, Eq. 17) and then use it explicitly as a forcing of the new experimental setup. For a more sophisticated forcing, a vertical dependency of the temperature is further considered via a lapse rate. Default setup in Nix accounts for adiabatic conditions $\Gamma_a = -9.8$ ºC/km, though a moist lapse rate is also available $\Gamma_m = -5.0$ ºC/km (e.g., Stone and Carlson, 1979).

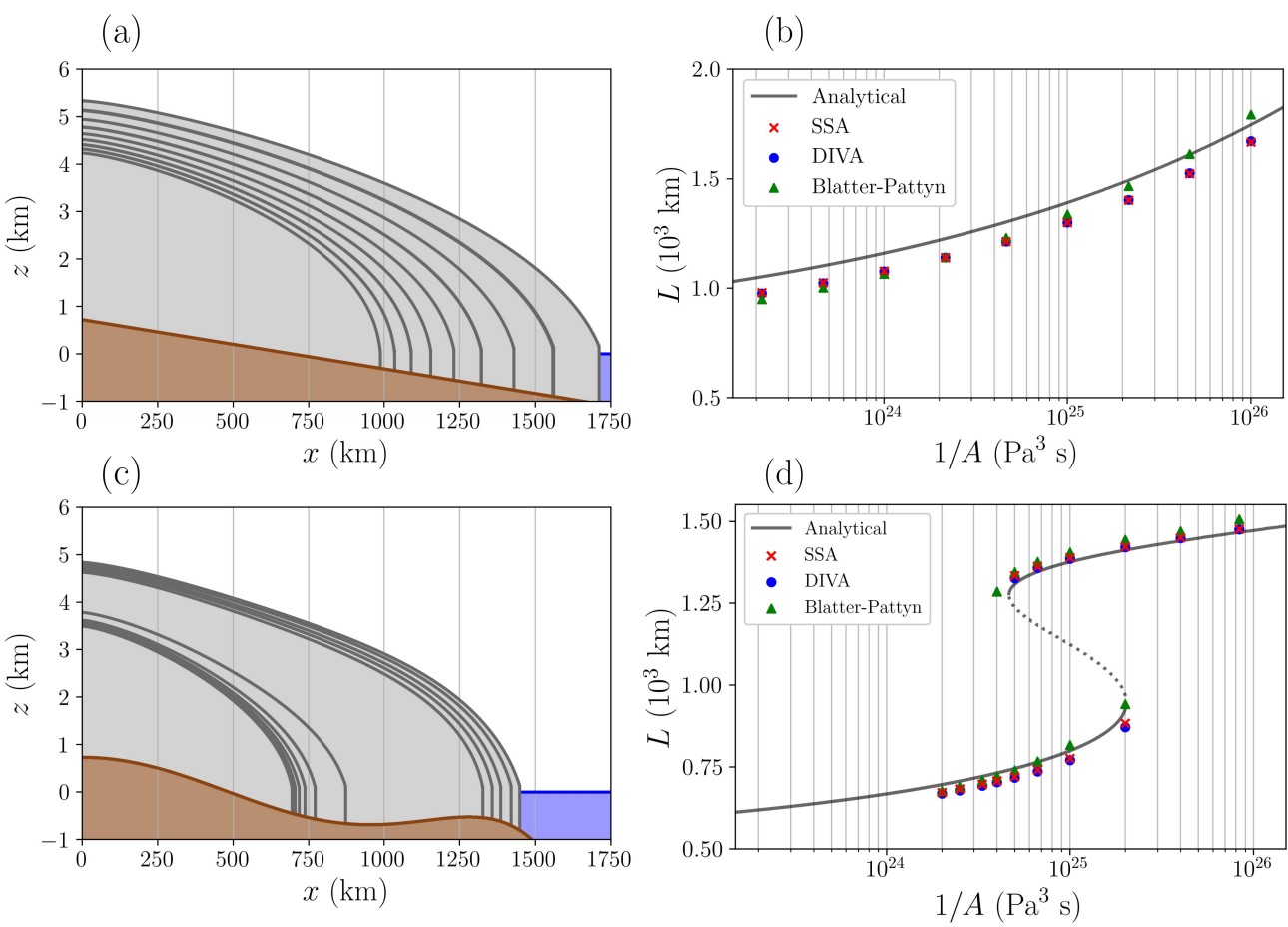

**Figure 3.** Left column: ice-sheet extent. Right column: grounding-line position as a function of the MISMIP forcing $A$ for three independent Stokes approximations: SSA, DIVA and Blatter-Pattyn. Grey line represents the analytical solution at equilibrium from Schoof (2007b): solid line, stable branch; dashed line, unstable branch. Markers represent Nix's results after the equilibration time given in Pattyn et al. (2012). Bed geometries correspond to: first row, Experiments 1 and 2 (both advance and retreat); second row, Experiment 3.

The particular atmospheric forcing is imposed at the sea level $T_{\mathrm{air}}(t)$ as shown in Fig. 4a. Starting from warm conditions $T_{\mathrm{air}} = 0$ºC, it reaches a minimum value of $-30$ºC in gradual steps that last 40 kyr each to ensure thermal quasi-equilibrium. Nonetheless, lower temperatures are present near the ice divide as the surface extends far above the sea level, wherein the lapse rate correction becomes relevant. This experiment reflects the insulating effect of the ice sheet as the forcing eventually reaches the initial atmospheric temperature but the grounding line does not retreat (Fig. 4b and 4f). It is not possible to make a one-to-one comparison with (Schoof, 2007b), given the different physical description of the system. Nonetheless, it is illustrative to represent the grounding-line position as a function of the ice temperature therein evaluated at two different depths and the vertical mean (Fig. 4b). The near-base temperature closely matches the theoretical prediction by the boundary layer. The lower layers appear to be shifted to the right, as the effect of the warmer surface becomes relevant.

In terms of the hysteresis behaviour, the jump over the retrograde region of the bed geometry occurs for near base temperature of $-30$ºC, as predicted by the semi-analytical counterpart (see grey solid line, Fig. 4). Even so, when the forcing returns to the initial value, the grounding line does not retreat back to its original position and remains advanced (black square in Fig. 4b). On the contrary, for shallower ice layers, the warming branch extends far from the analytical results, thus showing larger bistability against atmospheric temperature changes. As for the near-base layer, when the initial forcing state is eventually retrieved, the ice sheet extends beyond the bedrock peak and does not retreat. This is also well captured in Fig. 5 by comparing panels 5b and 5f knowing that both are equilibrium states with identical forcing. These results strongly differ when using a downwards-sloping bed geometry, as shown in panels 5a, 5c 5e, where thermodynamics is also active but not hysteresis is present as the bedrock does not present retrograde regions.

Lastly, it is worth noting the temperature proximity to melting point particularly at right hand side of the base in all panels due to the combined contribution of geothermal heatflux and frictional heat dissipation, Fig. 5a, 5b, 5e and 5f favoured by a warmer atmospheric temperature. Unlike panels 5c and 5d, where the lower surface temperature perturbs the entire temperature profile, thus only partially cooling the ice sheet base. It must be stressed that near the grounding line, there is a reduction in the basal temperature as a result of a considerably thinner ice, thus providing less thermal insulation of the colder surface.

### 6.2.2 Ocean temperature anomalies $\Delta T_{\mathrm{oce}}$ forcing

In this configuration, we apply ocean temperature anomalies with respect to a reference value $T_0$, so that $\Delta T_{\mathrm{oce}} = T - T_0$. We then convert these temperature anomalies into sub-shelf melting at the grounding line (e.g., Favier et al., 2019) by using any of the parametrizations shown in Section 3.8.

We first perform two identical hysteresis experiments forced by $\Delta T_{\mathrm{oce}}$ that solely differ on the thermodynamic treatment of the ice: an idealised fixed ice rate factor $A$ (Fig 6, blue curve) and a more realistic active thermodynamic scenario with a constant boundary condition $T_{\mathrm{air}}$ (Fig 6, red curve). Results show that active thermodynamics considerably widens the width of the hysteresis loop. This behaviour resembles that obtained for the atmospheric-forced simulations (Fig. 4), where we find a larger extent of the cooling branch compared to the semi-analytical solutions (grey line, Fig. 4).

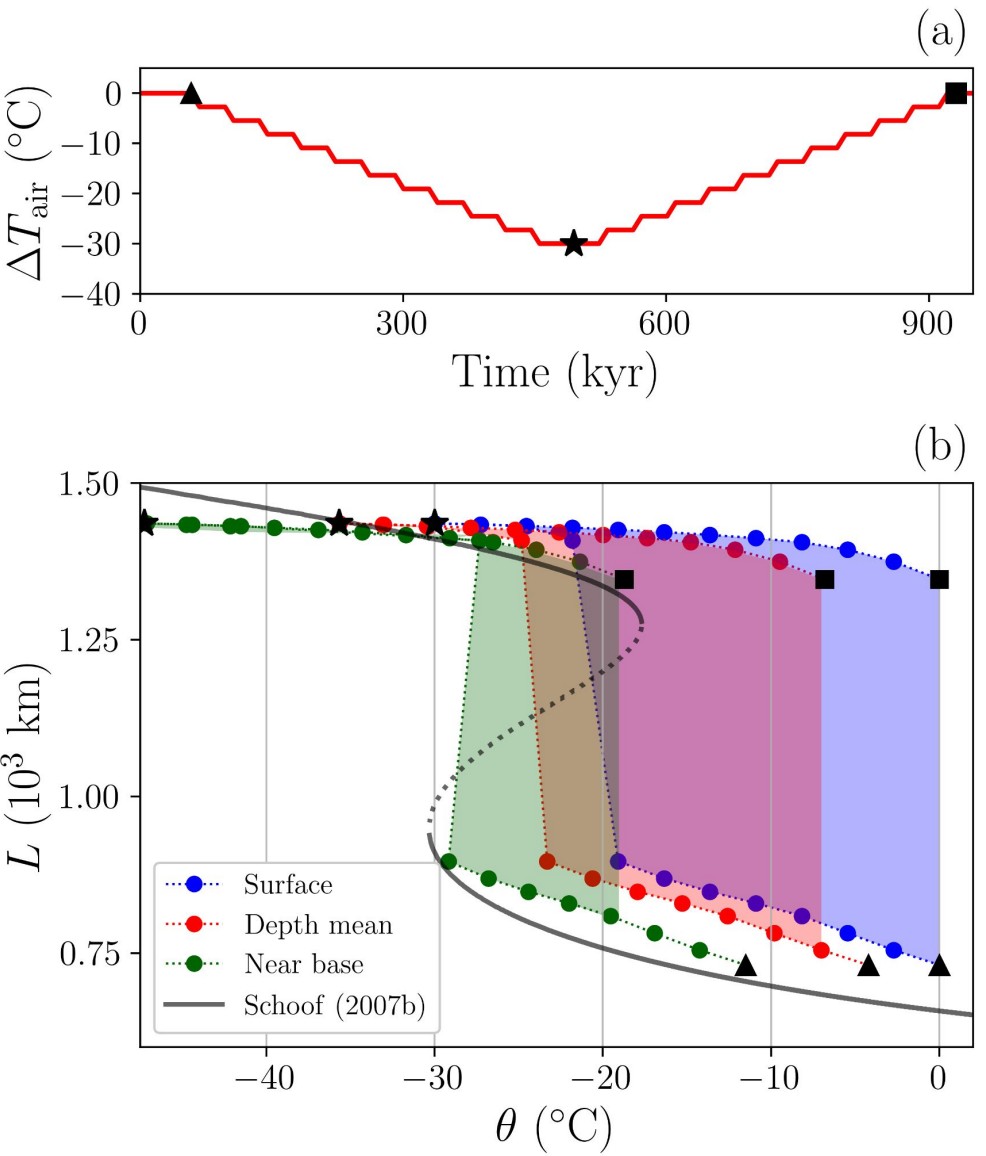

**Figure 4.** Overdeepened bed experiment forced with atmospheric temperatures. (a) Forcing time series: atmospheric temperatures $T_{\mathrm{air}}(t)$. Each black symbol represents three snapshots of particular interest: initial state (triangle, warmest conditions), coldest forcing conditions (star) and final state (square, same exact atmospheric conditions as the beginning). (b) Grounding line position as a function of the ice temperature evaluated at two different depths (near-base and surface) and vertical mean. Note that the initial and end states strongly differ in ice extent even though the atmospheric forcing is identical. The grey line represents analytical results in the absence of thermodynamics (i.e., imposed rate factor $A$), following Schoof (2007b).

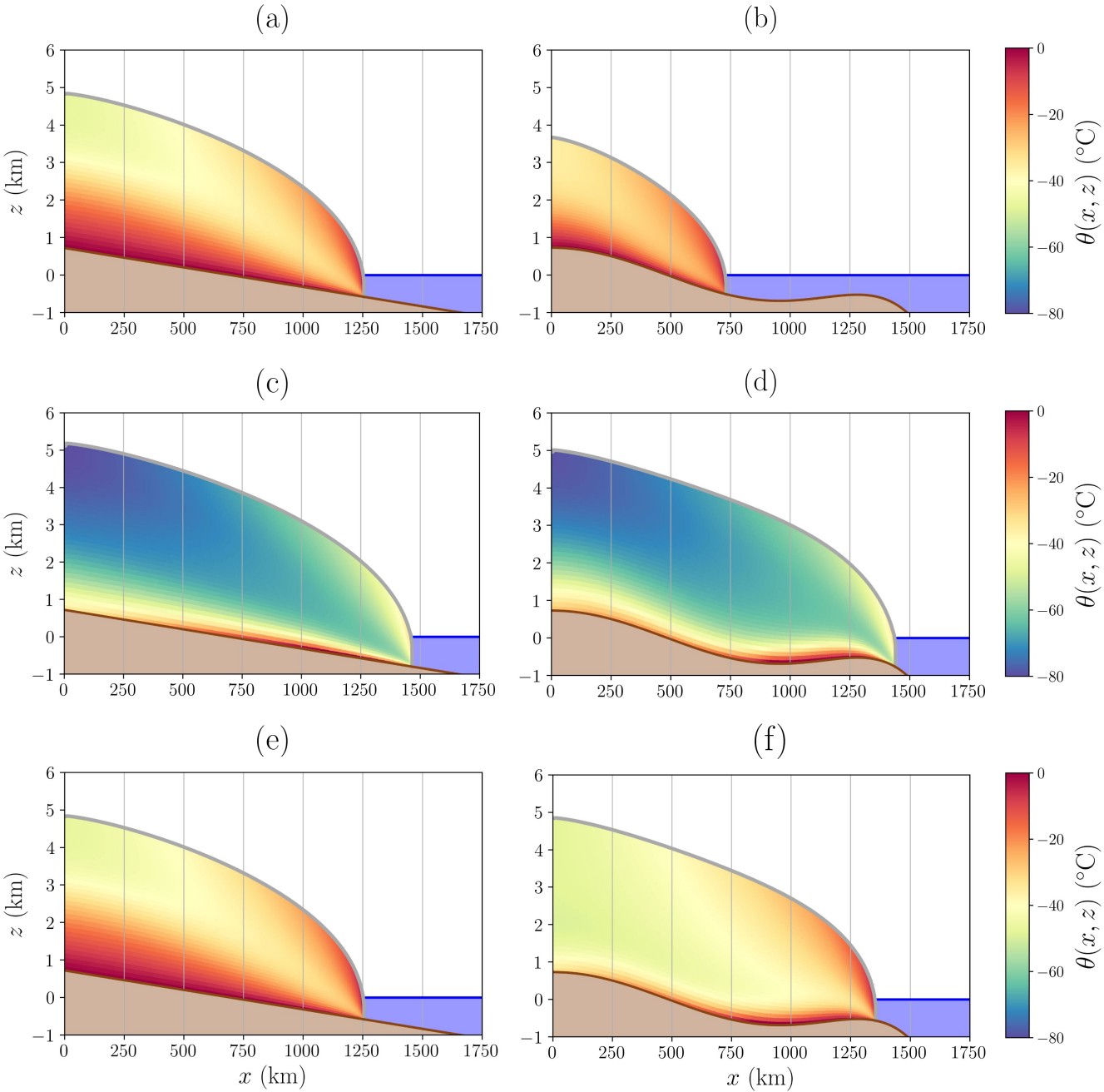

**Figure 5.** Ice sheet extent and temperature distribution for the prograde (left column) and the overdeepened (right column) bed geometries. Each row represents a snapshot given by each symbol in Fig. 4 (triangle, star and square, respectively): initial warm state (first row), coldest atmospheric conditions (second row) and final atmospheric configuration (third row). Note that the overdeepening bed geometry yields a final ice sheet profile extended far beyond the initial state even though the boundary conditions are identical, thus exhibiting hysteresis. Colours indicate the ice temperature at the given time.

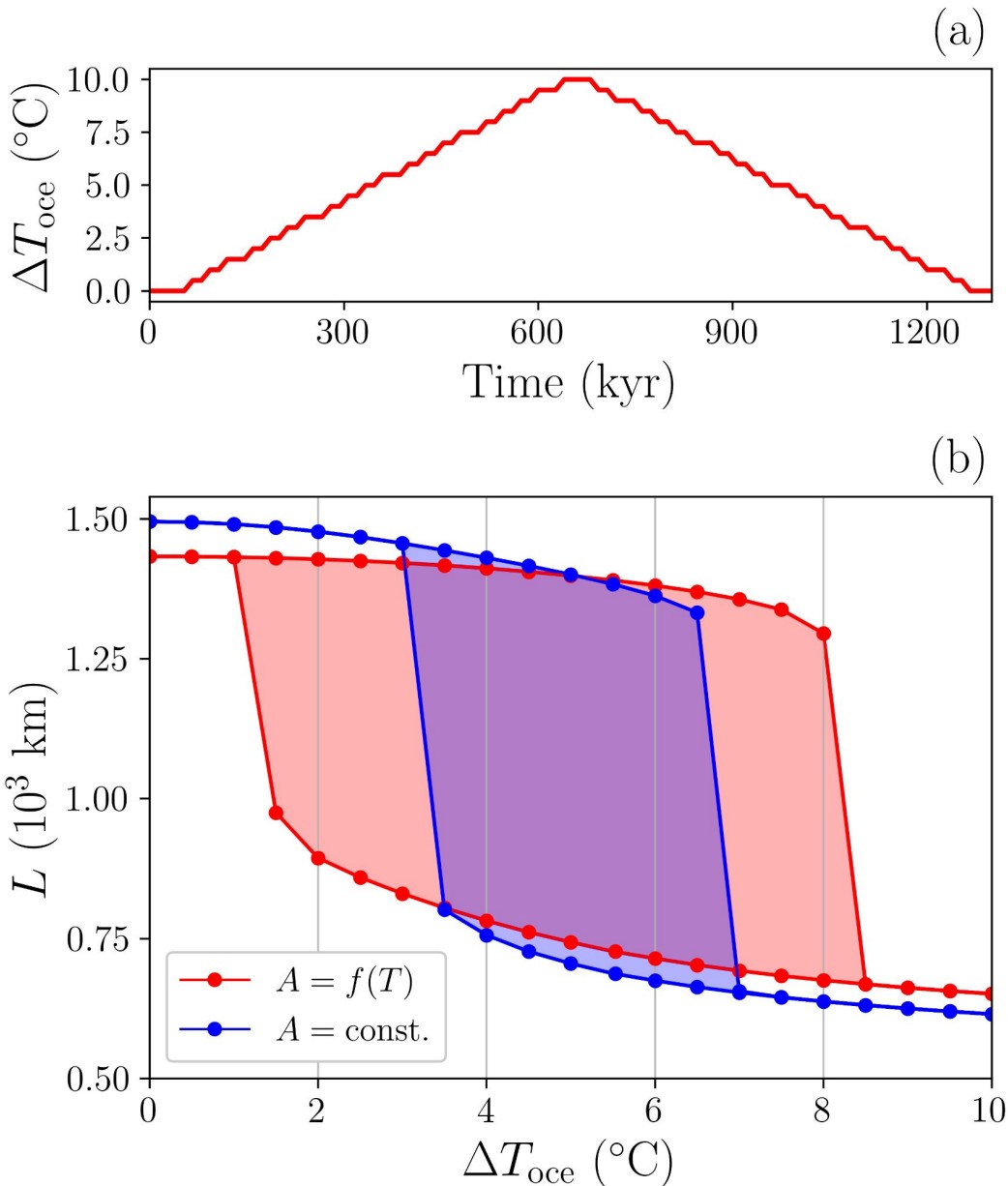

**Figure 6.** (a) External forcing time series: ocean temperature anomalies $\Delta T_{\mathrm{oce}}(t)$. Time duration of each step equals 30 kyr. (b) Hysteresis experiments for the overdeepened bed geometry forced via slowly-varying ocean temperature anomalies $\Delta T_{\mathrm{oce}}(t)$. Blue: constant ice rate factor $A = 10^{-26}$ Pa$^3$s. Red: active thermodynamics $A = f(T)$ with fixed boundary condition $T_{\mathrm{air}} = -40$ ℃. Each forcing step is ran for 30 kyr to ensure quasi-equilibrium (solid dots). A quadratic sub-shelf parametrisation is employed in both scenarios. Heat exchange velocity parameter $\gamma = 10^{-3}$ m/s.

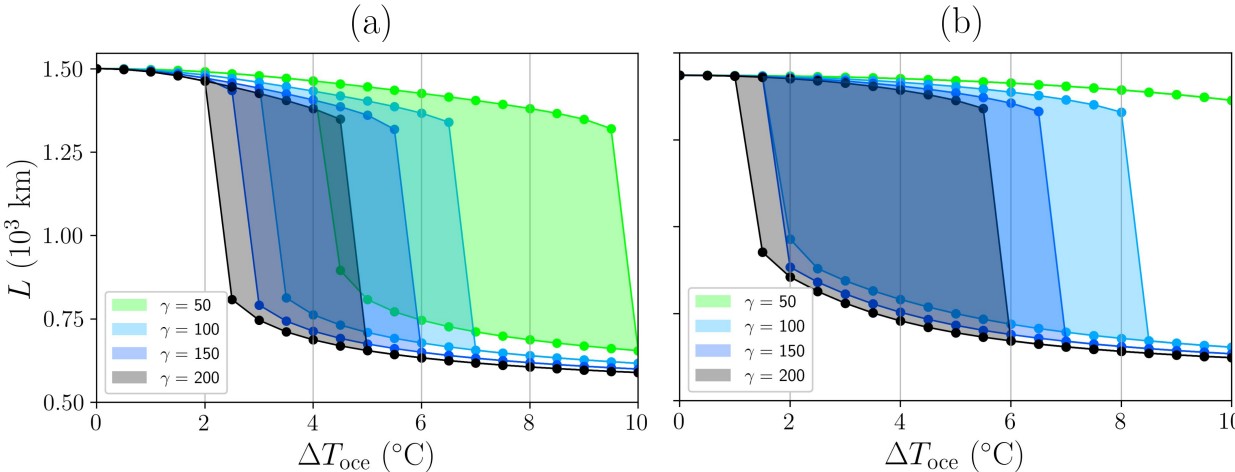

**Figure 7.** Sensitivity tests. As in Fig. 6a, but for (a) Constant ice rate factor $A = 10^{-26}$ Pa$^3$s. (b) Active thermodynamics with fixed $T_{\mathrm{air}}$. Values of the heat exchange velocity parameter $\gamma$ are given in $10^{-5}$ m/s and fall within the spanned range in Favier et al. (2019).

In addition to the experiments carried out to assess the importance of thermodynamics on the hysteresis behaviour of a marine
terminating ice sheet, we have performed a sensitivity study to quantify the differences caused by parameter uncertainty (e.g., Favier et al., 2019), particularly on the heat exchange velocity $\gamma$.

Figure 7 illustrates the high sensitivity to that stems from the heat exchange velocity parameter $\gamma$. It is worth noting that the retreat is much more sensitive to the particular $\gamma$ choice than the later advance as the anomalies approach zero. Namely, all intermediate values advance at $\Delta T_{\mathrm{oce}} = +1.5$ºC. On the contrary, the retreat occurs for a wider range of temperature anomalies
from +4.5ºC to +6.5ºC for $\gamma = 2.1 \times 10^{-5}$ m/s and $1.3 \times 10^{-5}$ m/s, respectively.

For a quadratic sub-shelf parametrisation (Eq. 22), the retreat takes place at $\Delta T_{\mathrm{oce}} = +2.5$ºC for the highest heat exchange velocity calibrated in Favier et al. (2019), i.e., $\gamma = 100 \times 10^{-5}$ m/s. Even the lowest parameter value presented in the same work also presents a retreat if the ocean temperature anomalies reach $\Delta T_{\mathrm{oce}} = +6.5$ºC. Multiple different values of $\gamma$ advance back at nearly the same particular forcing value $\Delta T_{\mathrm{oce}}$ 7, whereas the retreat happens at significantly different values. To
illustrate this, we can take the hysteresis loops corresponding to $\gamma = 20 \times 10^{-5}$ m/s, $25 \times 10^{-5}$ m/s, $35 \times 10^{-5}$ m/s. They respectively retreat at $\Delta T_{\mathrm{oce}} = +5.5$ºC, +5.0ºC and +4.5ºC, whereas the advance take place at precisely the same anomaly value $\Delta T_{\mathrm{oce}} = +2.5$ºC.

## 7  Discussion

The results of MISMIP benchmark experiments are successful given the good agreement between our numerical solution and
the semi-analytical work of Schoof (2007b) (Fig. 3). From a modeling perspective, our grounding-line position is slightly shifted upstream, like in other moving grid models shown in Pattyn et al. (2012). Nevertheless, a sensitivity test to spatial reso-

lution shows an asymptotic convergence towards the semi-analytical solution (Fig. H1, Appendix H), thus providing robustness to our results. A further comparison among the Nix velocity solvers show an excellent agreement on the equilibrium solutions for both bed geometries herein studied: downward sloping and overdeepening.

For active thermodynamics and air temperature forcing, the corresponding temperature range spanned by MISMIP ice rate factor $A(T)$ does not yield a full advance/retreat of the ice sheet. This can be understood by the insulating effect of the ice sheet. The MISMIP idealised forcing with varying rate factor simultaneously modifies the ice viscosity over the entire domain, whereas the real temperatures given by an active thermomechanical solver will adjust to the new surface temperature (Fig. 4). This further means that the impact on the viscosity is weaker as there are other heat sources as the basal friction dissipation

and the geothermal heat flow.

      The stability of the system is accordingly perturbed, as shown in Fig. 6. Particularly, the bistability of the system is increased, in the sense that a larger range of perturbation values have two stable solutions. To illustrate this, for a rate factor $A = 10^{-26}$ Pa$^3$s, the oceanic anomaly perturbation ranges from $\Delta T_{\mathrm{oce}} = +3$ to $+7$℃, whereas for the thermomechanically active scenario (i.e., temperature-dependent viscosity), the bistability solution is found for a wider range from a $\Delta T_{\mathrm{oce}} = +1.0$ to $+8.0$℃. This

can be understood as the result of a thermal adjustment that occurs when the temperature of the ice can evolve in time. As the forcing changes over time (i.e., the ocean temperature anomalies), the ice flux at the grounding line is modified and the inland ice thickness is perturbed accordingly. The new ice thickness distribution implies a slightly different temperature solution and the viscosity is consequently modified. Knowing that the viscosity field determines the velocity solution via the stress balance, we therefore find a clear feedback that allows for the ice sheet to adjust if thermodynamics are active.

It must be noted that this stability study employs a time duration in each forcing step of 30 kyr to elude transient responses (Fig.6b). This allows for the ice geometry to reach a steady-state as these experiments are intended to be quasi-equilibrium simulations (as MISMIP). If the time between steps was reduced, the hysteresis loop would then be a transitory response given that the ice temperature may not adjust to the new geometry. The added value of the thermomechanical coupling does not solely rely on the transitory response (i.e., thermal inertia), but also on the perturbed stability of the quasi-equilibrium hysteresis loop.

In other words, the thermomechanical coupling already determines the stable regions in a quasi-steady description and not only through the effects of thermal inertia. Our goal here is to first show this more fundamental behaviour. Further work is needed to assess the relevance of potential transient responses.

      It is worth noting the fundamental role of vertical advection if the system is to be forced with air temperatures. The magnitude of vertical advection and its vertical dependency determine the temperature distrubution within the ice (Moreno-Parada et al.,

2022). This mechanism strongly determines the equilibrum profiles as it modifies the overall outflow of ice via the viscosity dependency on temperatures (i.e., Arrhenius law Eq. 17).

      We must also discuss the calibrated values of $\gamma_T$ (Favier et al., 2019). The impact on the hysteresis behaviour is interesting as it does not imply a symmetric effect on the retreat/advance of the ice sheet (Fig. 7). Strictly speaking, the temperature anomaly necessary for the ice sheet to retreat is far more sensitive to the particular $\gamma$ value than the anomaly necessary to advance.

One potential explanation for this interesting behaviour rests on the very nature of the melting/calving parametrisation at the grounding line. Since $M$ grows with $\Delta T_{\mathrm{oce}}$ (Eq. 21 and 22), the difference among $M$ values for a fixed $\gamma$ increases with the

temperature anomaly. Hence, knowing that the advance occurs when the ice flux value reaches a certain value at the grounding line, the temperature anomaly range covered by different $\gamma$ that yields such a melting/calving values is smaller (as $\Delta T_{\mathrm{oce}} \to 0$). Moreover, if thermodynamics is considered (Fig. 7b), the required calving at the grounding line to retreat is generally larger

than that for a fixed rate factor (Fig. 7b). In fact, for sufficiently law values of $\gamma$, the thermomechanically active ice sheet never retreats. The thermal behaviour of the ice thus provides additional *inertia* in the sense that the ice sheet is less prone to change its current state, thus endowing the system with higher stability. This results is not exclusive of the oceanic forcing, as it also shown in Fig. 4, where the ice sheet does not collapse to a retreated position when the perturbation vanishes.

Lastly, results herein presented show that active thermodynamics perturbs the hysteresis loop and the overall stability of an

ice sheet. The particular grid over which the equations are discretised do not alter this behaviour, for that our results are not numerical artefacts of the chosen mesh. Therefore, we do not expect our results to change for a different grid discretisation. More precisely, we expect the same physical behaviour as long as the ice viscosity varies upon temperature changes, irrespective of the chosen grid. The exact grounding-line position may differ for a different grid, yet the physical mechanism underlying this mechanism remains unperturbed. It is thus expected that other models with distinct meshes will exhibit a similar mechanism

to Nix simulations.

## 8    Conclusions

The thermomechanically-coupled 2D model Nix has been presented and thoroughly described. There are a number of novelties compared to other two-dimensional models: a stress balance given by the Blatter-Pattyn approximation, a fully coupled thermodynamics solver, explicit calculation of the grounding line by a stretched coordinate system, stochastic boundary conditions

capability, adaptive time stepping and potential melting/calving at the grounding line. Nix allows the user to choose between explicit and implicit solvers for the main differential equations, while numerical stability is ensured by a staggered grid.

First, Nix's performance was tested by reproducing Experiments 1, 2 and 3 from MISMIP benchmarks (Pattyn et al., 2012). Results were further compared to semi-analytical solutions (Schoof, 2007b) yielding an excellent agreement for all Stokes approximation available in Nix: SSA, DIVA and Blatter-Pattyn. In general, our grounding line position slightly underestimates

the boundary layer results in Schoof (2007b), likewise all moving-grid models participating in MISMIP. The well-known hysteresis behaviour in Experiment 3 is also captured.

The complexity of the system was further increased by solving the associated heat problem. This allows us to investigate to which extent the hysteresis behavior under parameter variations is perturbed as a result of a temperature-dependent viscosity. In so doing, we designed two different suites of experiments regarding the forcing variables of the system: air temperatures

$T_{\mathrm{air}}$ and ocean temperature anomalies $\Delta T_{\mathrm{oce}}$.

When forcing with air temperatures, the hysteresis loop width is widened and the system exhibits larger bistability as the dynamics are perturbed via the ice viscosity dependency on temperature. This is a direct result of the insulating effect of the ice, thus preventing the colder atmospheric temperatures to travel downwards from the uppermost layers. In an idealised

overdeepened bed geometry, it is necessary to reach an air temperature of $-40$℃ at the sea level (provided an adiabatic lapse rate dependency with height) for the grounding line to advance beyond the bedrock local maximum.

If the system is instead forced by ocean temperature anomalies (i.e., melting/calving at the grounding line), we find that the hysteresis behaviour also persists. Notably, the ocean temperature anomaly at which the ice sheet retreats depends on the particular heat exchange parameter. Our results show that the quadratic parametrisation retreats at $\Delta T_{\text{oce}} = +8.0$℃ of temperature anomaly. The system advances back to its unperturbed state at $\Delta T_{\text{oce}} = +1.0$℃ for the quadratic parametrisation.

These results show that a temperature-dependent ice viscosity provides inertia and stability to the ice sheet, regardless of the particular external forcing applied (i.e., oceanic or atmospheric). Therefore, the thermal state of the ice must be accounted even for low-dimensional or conceptual models, particularly if the potential sudden retreat of an ice sheet is to be assessed.

*Author contributions.* Daniel Moreno-Parada built the entire model, ran all the simulations, analysed the results and wrote the paper. All other authors contributed to analyse the results and writing the paper.

*Financial support.* This research has been supported by the Spanish Ministry of Science and Innovation (project IceAge, grant no. PID2019-110714RA-100), the Ramón y Cajal Programme of the Spanish Ministry for Science, Innovation and Universities (grant no. RYC-2016-20587) and the European Commission, H2020 Research Infrastructures (TiPES, grant no. 820970).

*Competing interests.* The authors have no competing interests to declare.

*Code availability.* Nix ice-sheet model v1.0 is in a persistent Zenodo repository (Moreno-Parada et al., 2023): https://doi.org/10.5281/zenodo.10228874. Additionally, there is a GitHub repository where current and future versions of the software can be accessed at: https://github.com/d-morenop/nix

*Data availability.* TEXT

*Code and data availability.* TEXT

*Sample availability.* TEXT

## Appendix A: Discretization schemes and nonuniform grids

We herein elaborate on a thorough description of the discretization schemes of our flow line model where we follow the ordinary notation $q(\sigma_i, \zeta_j, \tau_n) \equiv q_{i,j}^n$.

The position in the spatio-temporal coordinates is then given by $\sigma_i = \sum_{k=0}^{i} \Delta\sigma_k$, $\zeta_i = \sum_{k=0}^{i} \Delta\zeta_k$ and $\tau_n = n\Delta\tau$. Note that Nix allows for a nonuniform spatial grid where the spacing between two consecutive points follows a desired distribution (polynomial or exponential). This yields high resolutions near the grounding line whilst minimising the total number of grid points. The horizontal index $i \in \mathcal{W}_r = \{0, 1, 2, ..., r\}$ where $r$ is the number of points in which the horizontal axis is divided. Likewise, the vertical index follows $j \in \mathcal{W}_p = \{0, 1, 2, ..., p\}$ where $p$ is the number of vertical layers.

Nix finite differences discretisation considers unevenly-spaced grids, commonly used in the glaciological community where higher resolutions are desired near the base whilst minimising the required number of points to reduce computational costs. A new coordinate system is thus built considering two types of nonuniform grid spacing: polynomial and exponential. For any normalised variable $\xi$ (such as $\sigma$ and $\zeta$, Eq. 23), a new nonuniform grid $\tilde{\xi}$ can be obtained by a power law transformation:

$$\tilde{\xi} = \xi^n \tag{A1}$$

where $n$ is the spacing order, or an exponential map:

$$\tilde{\xi} = \frac{e^{s\xi} - 1}{e^s - 1} \tag{A2}$$

where $s$ is the spacing factor for the exponential grid and $\tilde{\xi}$ is substituted by the variables $\sigma$ and $\zeta$. In this study, we have employed $n = 2$ and $s = 2$.

## Appendix B: Blatter-Pattyn stress balance discretization.

The discretization is straightforward for an Arakawa-C grid. The position of the grounding line $L(t)$ is located on the velocity grid (following Vieli and Payne, 2005). Thus, if the horizontal axis is divided in $r$ points, the ice thickness grid ranges $i = 0, 1, ..., r-1$, whereas the velocity grid (staggered) indexes read $i = 1/2, 3/2, ..., r-1/2$. The fractional index implies that the point $(i+1/2, j)$ lies between $(i, j)$ and $(i+1, j)$ and analogously for the vertical index $j$.

The Blatter-Pattyn stress balance can be written as:

$$\frac{2}{L^2 \Delta\sigma_{i+1/2}} \left[ \eta_{i+1,j} \frac{u_{i+3/2,j} - u_{i+1/2,j}}{\Delta\sigma_{i+3/2} + \Delta\sigma_{i+1/2}} - \eta_{i,j} \frac{u_{i+1/2,j} - u_{i-1/2,j}}{\Delta\sigma_{i+1/2} + \Delta\sigma_{i-1/2}} \right] +$$
$$\frac{1}{2(H_i)^2 \Delta\zeta_{j+1/2}} \left[ \eta_{i,j+1} \frac{u_{i,j+3/2} - u_{i,j+1/2}}{\Delta\zeta_{j+3/2} + \Delta\zeta_{j+1/2}} - \eta_{i,j} \frac{u_{i,j+1/2} - u_{i,j-1/2}}{\Delta\zeta_{j+1/2} + \Delta\zeta_{j-1/2}} \right] = \rho g \frac{h_{i+1} - h_i}{L\Delta\sigma_{i+1/2}}, \tag{B1}$$

We thus have a linear system of $r \times p$ unknowns that can be solved applying standard linear algebraic solvers. For each timestep, we build a matrix of coefficients with dimension $(rp) \times (rp)$:

$$\underset{(rp)\times(rp)}{A} * \underset{(rp)\times(1)}{u} = \underset{(rp)\times(1)}{F} \tag{B2}$$

Since our discretization stencil includes 6 points: $(i+3/2, j)$, $(i+1/2, j)$, $(i-1/2, j+1)$, $(i, j+3/2)$, $(i, j+1/2)$ and $(i, j-1/2)$, we obtain a sparse matrix that allows for optimised inversion. For $r = 500$ and $p = 25$, only a $0.048\%$ of the

coefficient matrix are nonzero entries:

$$
\begin{pmatrix}
 & & & & & & & & \\
 & & & & & & & & \\
\alpha_{i-2M,j} & \cdots & \alpha_{i-M,j} & \cdots & \alpha_{i,j-1/2} & \alpha_{i,j+1/2} & \cdots & \alpha_{i+M,j}\cdots & \alpha_{i+2M,j} \\
 & & & & & & & & \\
 & & & & & & & & \\
\end{pmatrix}
\begin{pmatrix}
u_{i-3/2,j} \\ \vdots \\ u_{i-1/2,j} \\ \vdots \\ u_{i,j-1/2} \\ u_{i,j+1/2} \\ \vdots \\ u_{i+1/2,j} \\ \vdots \\ u_{i+3/2,j}
\end{pmatrix}
=
\begin{pmatrix}
F_{i-3/2,j} \\ \vdots \\ F_{i-1/2,j} \\ \vdots \\ F_{i,j-1/2} \\ F_{i,j+1/2} \\ \vdots \\ F_{i+1/2,j} \\ \vdots \\ F_{i+3/2,j}
\end{pmatrix}
\tag{B3}
$$

The boundary conditions are then needed to evaluate the velocity at the edge of the domain. For the free surface we apply a three-point derivative scheme:

$$u_{i,r-1/2} = \frac{4u_{i,r-3/2} - u_{i,r-5/2} + 2\chi\Delta\zeta_{r-3/2}}{3}. \tag{B4}$$

Where $\xi$ is evaluated right below the uppermost layer (i.e., $j = r - 3/2$):

$$\chi = 4\frac{u_{i+1/2,r-3/2} - u_{i-1/2,r-3/2}}{\Delta\sigma_{i+1/2} + \Delta\sigma_{i-1/2}} \frac{h_{i+1} - h_i}{\Delta\sigma_{i+1/2}}. \tag{B5}$$

The basal boundary condition is thus obtained analogously with a non-zero drag. In this case, we discretise with a simple 2-point scheme:

$$u_{i,1/2} = u_{i,3/2} - \mu\Delta\zeta_{i,1/2}. \tag{B6}$$

Where $\mu$ is computed right above the basal boundary (i.e., $j = 3/2$):

$$\mu = 4\frac{u_{i+1/2,3/2} - u_{i-1/2,3/2}}{\Delta\sigma_{i+1/2} + \Delta\sigma_{i-1/2}} \frac{b_{i+1} - b_i}{\Delta\sigma_{i+1/2}}. \tag{B7}$$

With these expressions, the velocity is field is then complete. Lastly, at the ice divide and the grounding line, the velocity is obtained from symmetry and hydrostatic equilibirum arguments, respectively (elaborated in Appendix C).

## Appendix C: DIVA/SSA stress balance discretization.

The discretization is straightforward for a staggered grid. The position of the grounding line $L(t)$ is located on the staggered grid (following Vieli and Payne, 2005). Thus, if the domain is divided in $n$ points, the ice thickness grid ranges $i = 0, 1, ..., r-1$, whereas the velocity grid (staggered) indexes read $i = 1/2, \, 3/2, ..., r-1/2$.

The SSA stress balance can be then written as (note that the factor 2 difference in Eq. B.15, Vieli and Payne, 2005 since their viscosity definition is not preceded by $1/2$ is cancelled by the average between two consecutive grid lengths $\Delta\sigma$ necessary to compute the velocity gradients):

$$\frac{2}{L^2\Delta\sigma_{i+1/2}} \left[ \eta_{i+1}H_{i+1}\frac{u_{i+3/2,j} - u_{i+1/2,j}}{\Delta\sigma_{i+3/2} + \Delta\sigma_{i+1/2}} - \eta_i H_i \frac{u_{i+1/2,j} - u_{i-1/2,j}}{\Delta\sigma_{i+1/2} + \Delta\sigma_{i-1/2}} \right]$$
$$-\beta_{i+1/2}^2 u_{i+1/2} = \rho g \frac{H_{i+1} + H_i}{2}\frac{h_{i+1} - h_i}{L\Delta\sigma_{i+1/2}}, \tag{C1}$$

where the friction coefficient $\beta^2$ reads:

$$\beta_{i+1/2} = Cu_{i+1/2}^{m-1}, \tag{C2}$$

so that $\tau_b = \beta^2 u$.

Assuming our domain is divided in $n$ points, the corresponding tridiagonal matrix is built at every time step as (where we have dropped the super-index to lighten the notation):

$$\begin{pmatrix} B_0 & C_0 & & & \\ A_1 & B_1 & C_1 & & \\ & A_2 & B_2 & \ddots & \\ & & \ddots & \ddots & C_{r-2} \\ & & & A_{r-1} & B_{r-1} \end{pmatrix} \begin{pmatrix} u_{1/2} \\ u_{3/2} \\ \vdots \\ u_{r-3/2} \\ u_{r-1/2} \end{pmatrix} = \begin{pmatrix} F_0 \\ F_1 \\ \vdots \\ F_{r-2} \\ F_{r-1} \end{pmatrix} \tag{C3}$$

where $A_0 = C_{n-1} = 0$ by definition.

The non-zero entries of the matrix and the inhomogeneous term read:

$$A_i = \gamma_i \eta_i H_i$$
$$B_i = -\gamma_i \left( \eta_{i+1}H_{i+1} + \eta_i H_i \right) - \beta_i^2$$
$$C_i = \gamma_i \eta_{i+1} H_{i+1}$$
$$F_i = \rho g \frac{H_{i+1} + H_i}{2}\frac{h_{i+1} - h_i}{L\Delta\sigma_{i+1/2}} \tag{C4}$$

where $\gamma_i = 2/(L\Delta\sigma_{i+1/2})^2$.

For the edge of the matrix (i.e., $i = n - 1$), we use the following values:

$$A_{r-1} = \eta_{r-1} H_{r-1}$$

$$B_{r-1} = -\gamma_{r-1} \left( \eta_{r-1} H_{r-1} \right) - \beta_{n-1}^2$$

$$C_{r-1} = 0$$

$$F_{r-1} = \rho g H_{r-1} \frac{h_{r-1} - h_{r-2}}{L \Delta \sigma_{r-3/2}} \tag{C5}$$

For the boundary values, we set (note that, in the staggered grid, $u_{1/2}$ is the very first velocity value of the domain):

$$u_{1/2} = -u_{3/2},$$

$$u_{r-1/2} = u_{r-3/2} + \frac{L \Delta \sigma_{r-3/2}}{8 \eta_{r-1}} \left( \rho g H_{r-1}^2 - \rho_w g D^2 \right) \tag{C6}$$

where $D$ is the bedrock depth below sea level, the first equality yields from symmetry arguments at the ice divide ($i = 1$) and the second implies that the ice momentum is equated by the hydrostatic pressure of the water.

## Appendix D: Advection discretization

For the advection equation we also chose an implicit scheme for numerical stability:

$$\frac{H_i^{n+1} - H_i^n}{\Delta \tau^n} = \sigma_i \frac{\dot{L}^n}{L^n} \frac{H_{i+1}^{n+1} - H_{i-1}^{n+1}}{\left( \Delta \sigma_{i+1/2} + \Delta \sigma_{i-1/2} \right)} - \frac{2 \left( q_{i+1/2}^{n+1} - q_{i-1/2}^{n+1} \right)}{L^n \left( \Delta \sigma_{i+1/2} + \Delta \sigma_{i-1/2} \right)} + S_i^n, \tag{D1}$$

where the ice flux is defined as:

$$q_{i+1/2} = u_{i+1/2} \frac{H_{i+1} + H_i}{2} \tag{D2}$$

However, at the grounding line we use:

$$q_{r-1/2} = u_{r-1/2} H_{r-1} + M, \tag{D3}$$

where $M$ represents a flux anomaly at the terminus driven by variable ocean conditions. In real glaciers, these anomalies could be driven by variable calving, submarine melt, or a combination.

The advection equation can be rewritten as:

$$A_i H_{i-1}^{n+1} + B_i H_i^{n+1} + C_i H_{i+1}^{n+1} = F_i \tag{D4}$$

so that the corresponding matrix is also tridiagonal:

$$A_i = \gamma_i^n \left( \sigma_i \dot{L} - u_{i-1} \right)$$

$$B_i = 1 + \gamma_i^n \left( u_i - u_{i-1} \right)$$

$$C_i = \gamma_i^n \left( -\sigma_i \dot{L} + u_i \right)$$

$$F_i = H_i^n + \Delta \tau^n S_i^n \tag{D5}$$

where $\gamma_i^n = \Delta\tau^n / 2 \left( \Delta\sigma_{i+1/2} + \Delta\sigma_{i-1/2} \right) L^n$.

As the ice divide is located at $i = 1$ (note that we start counting at $i = 0$), the boundary condition then reads:

$$H_0^n = H_2^n, \tag{D6}$$

since $\sigma = 1$ is a symmetry axis.

## Appendix E: Grounding line scheme

The terminus position L (i.e., the grounding line) is not fixed in time. Direct discretization of Eq. 18 in terms of $\sigma$-coordinates leads to:

$$\dot{L}^n \equiv \frac{dL}{d\tau} = \frac{-L^n \Delta\sigma_{r-1/2} S_{r-1}^n + 2 \left( q_{r-1/2}^n - q_{r-3/2}^n \right) / \left( \Delta\sigma_{r-1/2} + \Delta\sigma_{r-3/2} \right)}{H_{r-1}^n - H_{r-2}^n + \varrho \left( b_{r-1}^n - b_{r-2}^n \right)}, \tag{E1}$$

## Appendix F: Thermodynamics discretization scheme

Unlike previous discretizations, the temperature field $\theta(\sigma, \zeta, \tau)$ has an additional dependency on the vertical coordinate $\zeta$ that brings a higher degree of complexity (Eq. 24).

The energy balance (Eq. 28) is discretized using an upwind scheme with a forward Euler step and a centred difference for the spatial derivatives. The lengthy expression reads:

$$\rho c \left[ \frac{\theta_{i,j}^{n+1} - \theta_{i,j}^n}{\Delta\tau^n} - \sigma_i \frac{\dot{L}^n}{L^n} \frac{\theta_{i+1,j}^n - \theta_{i-1,j}^n}{\Delta\sigma_{i+1/2} + \Delta\sigma_{i-1/2}} + \right.$$
$$\left. - \frac{\zeta_{i,j}}{H_i^n} \frac{H_i^{n+1} - H_i^{n-1}}{2\Delta\tau^n} \frac{\theta_{i,j+1}^n - \theta_{i,j-1}^n}{\Delta\zeta_{i,j+1/2} + \Delta\zeta_{i,j-1/2}} \right] = \frac{k}{(H_i^n)^2} \frac{\theta_{i,j+1}^n - 2\theta_{i,j}^n + \theta_{i,j-1}^n}{\left( \Delta\zeta_{i,j+1/2} + \Delta\zeta_{i,j-1/2} \right)^2} +$$
$$- \rho c \frac{u_i^n}{L^n} \left[ \frac{\theta_{i+1,j}^n - \theta_{i-1,j}^n}{\Delta\sigma_{i+1/2} + \Delta\sigma_{i-1/2}} - \left( \frac{b_{i+1} - b_{i-1}}{\Delta\sigma_{i+1/2} + \Delta\sigma_{i-1/2}} + \zeta_{i,j} \frac{H_{i+1}^n - H_{i-1}^n}{\Delta\sigma_{i+1/2} + \Delta\sigma_{i-1/2}} \right) \frac{\theta_{i,j+1}^n - \theta_{i,j-1}^n}{H_i^n \left( \Delta\zeta_{i,j+1/2} + \Delta\zeta_{i,j-1/2} \right)} \right] + \Phi_i^n \tag{F1}$$

## Appendix G: Adaptive time stepping

We take an adaptive timestepping approach to enhance the computational performance of the flowline model. Unlike the proportional-integral (PI) methods, we exploit the fact that Picard's iteration already computes a metric to determine convergence. Thus, without additional calculations, we let the new timestep to evolve within a range set by the user $[\Delta t_{\min}, \Delta t_{\max}]$ with a quadratic dependency on the error:

$$\Delta\tilde{t} = \left( 1 - \left( \frac{\min\left[ \varepsilon(t), \phi_{\text{pic}} \right]}{\phi_{\text{pic}}} \right)^2 \right) \left( \Delta t_{\max} - \Delta t_{\min} \right) + \Delta t_{\min}, \tag{G1}$$

where $\phi_{\text{pic}}$ is the tolerance on Picard's iteration and $\varepsilon(t)$ is the error on the current iteration defined as $\varepsilon^i = ||u^i - u^{i-1}||$ (Smedt et al., 2010). If the solution has not converged in the given timestep (i.e., $\varepsilon > \phi_{\text{pic}}$), Eq. Equation G1 ensures that the timestep is set to the minimum value.

Then, we apply some relaxation to provide stability and avoid spurious oscillations:

$$\Delta t = \alpha \Delta t + (1 - \alpha) \Delta \tilde{t}, \tag{G2}$$

where $\alpha = 0.7$. And we finally ensure that the timestep is no greater than the CFL condition:

$$\Delta t = \min [\Delta t, \Delta t_{\text{CFL}}] \tag{G3}$$

## Appendix H: Convergence and computational speed

It is challenging to give a one-to-one comparison since other models that solve for the higher-order momentum balance coupled
with a thermomechanical solver are full 3D solvers (partially providing Nix novelty). To give an estimation, MALI ice-sheet model (Hoffman et al., 2018) control simulations averaged 5.26 simulated years per wall-clock hour. On the contrary, MISMIP experiments run with Nix reach $\sim 10^5$ simulated years per wall-clock hour on average. Thus, there is a 5-order magnitude difference in terms of computational time.

Figure H1 illustrates the dependency of computational speed on the total number of horizontal grid points. The SSA and
685 DIVA solvers exhibit a similar performance for all resolutions, whereas the higher-order Blatter-Pattyn solver is nearly 2 orders of magnitude slower. This is simply a result of the inversion of the corresponding sparse matrix in the latter, significantly slower than the tridiagonal solver applied for both SSA and DIVA.

*Author contributions.* TEXT

*Competing interests.* TEXT

*Disclaimer.* TEXT

*Financial support.* This research has been supported by the Spanish Ministry of Science and Innovation (project Marine, grant no. PID2020-117768RB-I00). This research is TiPES contribution no. XXX and has been supported by the European Union Horizon 2020 research and innovation programme (grant no. 820970), the Ramón y Cajal Programme of the Spanish Ministry for Science, Innovation and Universities (grant no. RYC-2016-20587). Alexander Robinson received funding from the European Union (ERC, FORCLIMA, 101044247).

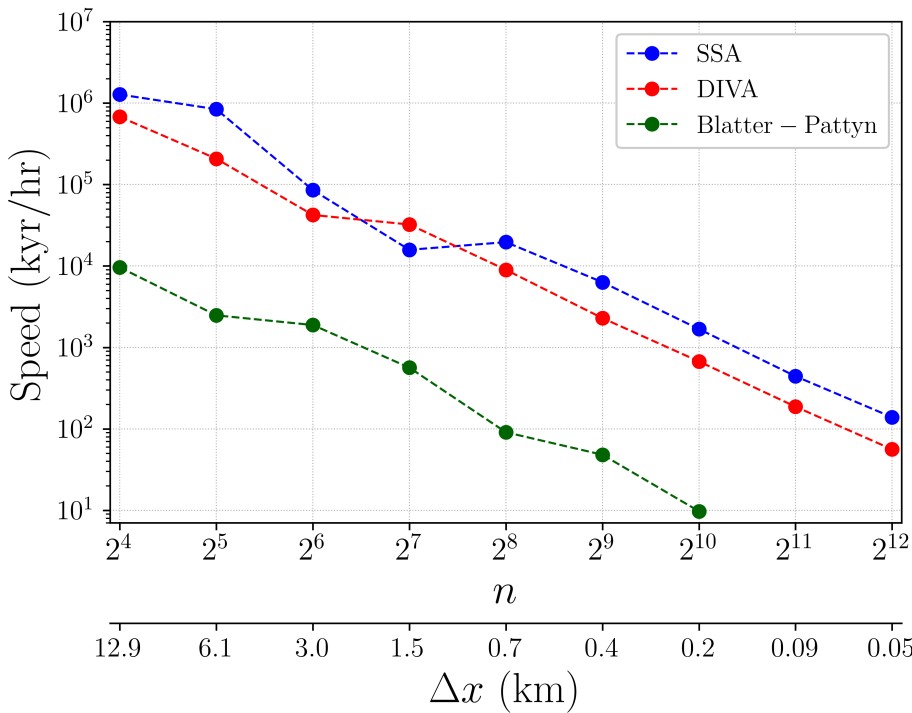

**Figure H1.** Nix computational speed for the three solvers available. The double $x$-axis represents the number of horizontal grid points $n$ and the corresponding resolution at the grounding line $\Delta x$. Note that Nix allows for an unevenly-spaced stretched grid that explicitly tracks the grounding line position.

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
