# Peer review of "Description and validation of the ice sheet model Nix v1.0"

_EGUsphere, 2023_

## Author Comment (AC1)

**Author's response to Anonymous Referee 1 (RC1) egusphere-2023-2690**

Daniel Moreno-Parada, Alexander Robinson,
Marisa Montoya, and Jorge Alvarez-Solas.

March 12, 2024

**1 Review**

The authors are very grateful to the reviewer for their constructive comments. We now provide our answers (regular font) to the main concerns raised by the reviewer (italic).

- *The fact that Nix is a 2D model significantly limits its potential as a community code, in my opinion. Although the model is well-described, which is expected for GMD, all components (physical models, numerics) employ well-known techniques that are already implemented in many other community glacier/ice sheet models in 3D (e.g., PISM, Elmer/Ice, CESM, ISSM, Bisicles, to name a few). Without offering anything distinctly new, I question the rationale behind publishing a 'model development' paper on the Nix model.*

We believe that the present work strongly adheres to those defined by "*Model description papers*" (in agreement with Referee #2, Tijn Berends), for that it meets the Geoscientific Model Development journal definition of a "*detailed, complete, rigorous and accessible description of a numerical model*" (see `https://www.geoscientific-model-development.net/`). It is well known that different models behave distinctly, and it is thus important to document them. Information about the real world is not only obtained from spatially explicit models and simpler models allow the investigation of the relevance of specific processes. Even though Nix model has a number of novelties (elaborated below), it should not be a requirement to be published as a "*Model description papers*" in GMD.

The main goal of this work is to present and describe an ice-sheet model that aims at overcoming the inevitable trade-off between a sophisticated physical description and fast computations. Compared to other 2D models, Nix provides the following novelties (summarised in Table 1):

1. **Thermomechanical coupling**. Nix explicitly resolves the energy balance by solving for the ice temperatures assuming a number of processes in the heat equation: vertical diffusion, horizontal and vertical advection, strain dissipation and basal frictional heat. As a result, the ice viscosity is thus temperature dependent. Moreover, the basal friction coefficient is also coupled to the thermal state of the base, thus accounting for a potential friction reduction if the pressure melting point is reached.

2. **Fast computations**. It combines a higher-order stress balance with active thermo-dynamics, while keeping extremely fast computations that allow for statistical studies with thousands of simulations involved, otherwise prohibited by 3D models.

   2a. For wall-clock times of the order of minutes, Nix allows for resolutions of $\Delta x = 0.1$ km (needed to properly resolve the grounding line) and simulated times of order

$t \sim 10^3$ kyr. This extremely low computational cost allows for statistical studies with thousands of simulations involved, otherwise prohibited by 3D models.

**2b.** Parallelisation. Nix users can optionally select parallel computing (supported by Eigen library), particularly convenient for high resolutions in the Blatter-Pattyn approximation, where large sparse matrices must be inverted. Moreover, it is possible to use Eigen's matrices, vectors, and arrays for fixed size within CUDA kernels.

**2c.** It is hard to give a one-to-one comparison since other models that solve for the same higher-order momentum balance coupled with a thermodynamical solver use 3D solvers (partially providing Nix novelty). To give an estimation, the MALI ice-sheet model (Hoffman et al., 2018) control simulations averaged 5.26 simulated years per wall-clock hour. On the contrary, MISMIP experiments run with Nix reach $\sim 10^5$ simulated years per wall-clock hour on average. Thus, there is a 5-order magnitude difference in terms of computational time.

3. **Time stepping**. Unlike other models, Nix offers an adaptive time stepping based on the convergence of Picard's iteration in the velocity solution solver. This approach differs from the standard proportional-integral (PI) methods (e.g., Cheng et al., 2017) and strongly contributes to its fast computational performance in Nix.

4. **Friendly usage**. As a combination of low computational demands, a 2D setup and a clean structure, Nix can be even run on a regular laptop. Installation, compilation and execution are controlled from a simple Python program. This allows Nix to be used without deep knowledge of the C programming language (low-level, procedural and statically-typed). Even though Nix simulations can run on a personal computer, the user can exploit parallelization on a High Performance Computing cluster.

Table 1: Summary of Nix capabilities.

| Requirement | Nix capability |
| --- | --- |
| Fast computations | Low dimensionality, adaptive time-stepping, parallelization |
| Higher-order description | Blatter-Pattyn model |
| Thermomechanical coupling | Temperature-dependent ice viscosity and basal friction |
| Easy usage | Python wrapper |

These capabilities constitute a notable difference from other ice-sheet models. Even so, we are fully aware of the limitations of a 2D setup. We do not aim at describing a highly sophisticated 3D ice sheet and we are thus conscious of the narrower range of possible applications of a 2D set-up, but we believe our model can provide additional insight for more comprehensive model studies. Two-dimensional models have tremendously helped in understanding ice-sheet dynamics both from a theoretical (e.g., Schoof, 2005; Schoof,

2006; 2007a; b) and a modelling perspective (e.g., Payne, 1995; Hindmarsh and Le Meur, 2001; Vieli and Payne, 2005; Pattyn et al., 2006). Moreover, 2D models are also capable of providing valuable insight on mechanisms such as buttressing (Dupont and Alley, 2005; Schoof, 2007; Jamieson et al., 2012, Robel et al., 2014; 2018). We have included a statement of the clear intended applications of Nix.

- *On the other hand, the MISMIP experiments incorporating active thermo-mechanics in response to oceanic and atmospheric forcings seem novel to me (has this been done before?). However, the findings are not thoroughly discussed, particularly in terms of connecting with existing literature, and the implications are barely addressed.*

To the authors' knowledge, this approach is indeed novel. We have further exploited these results, thus expanding the connection of our findings with existing literature. Moreover, we have addressed the potential implication of the results better in the discussion section. This will be incorporated into the new version of the manuscript.

- *In conclusion, I believe the originality of the manuscript resides more on its application than in model development. Therefore, I recommend shifting the focus of the paper to highlight the results more prominently, relegating the thorough yet unoriginal model description to an extensive appendix. By providing deeper discussions and a clearer emphasis on the results, a submission to a more applied journal, such as the Journal of Glaciology, The Cryosphere, or Frontiers, would be more fitting. I hope my comments are useful.*

As mentioned above, Nix's novelty lies in its fast computational capabilities and a low memory demand, thus allowing for high resolutions while explicitly solving the associated heat problem. In a broader framework of spatially explicit ice-sheet models, it can potentially provide important insight into grounding line dynamics in comprehensive 3D models. In addition, unlike other 2D models, Nix provides full thermomechanical coupling with both ice flow and basal friction.

Several of Nix's capabilities thus constitute a notable difference from other ice-sheet models and we are inclined to think that our model can provide valuable insight for more comprehensive studies. Nonetheless, we agree that our application of the model to the MISMIP experiments with coupled thermodynamics constitute interesting and novel results.

The corresponding analysis in the revised text has been consequently expanded. We believe that the manuscript now represents a more valuable contribution in this way.

---

## Author Comment (AC2)

**Author's response to Tijn Berends (RC2)**
**egusphere-2023-2690**

Daniel Moreno-Parada, Alexander Robinson,
Marisa Montoya, and Jorge Alvarez-Solas.

March 12, 2024

**1 Review**

The authors are deeply grateful to the reviewer for their constructive comments. We now provide our answers (regular font) to the main concerns raised by the reviewer (italic).

**Major comments**

- *Applicability of the 2-D set-up. I am not entirely convinced of the practical use of the "flowline+vertical" set-up presented here. I understand that such a model could help provide accurate benchmark solutions for future idealised-geometry experiments, but the fact that it can never be applied to a realistic ice sheet severely limits its range of possible applications. I therefore think the manuscript would benefit from a clear statement of the intended applications of the model. I am also unsure whether the term "ice-sheet model" is fitting for a model that cannot be used to study realistic ice sheets. Perhaps something like "idealised ice-sheet model" would be more fitting.*

We thank the reviewer for the opportunity to expand on the applicability of 2D models in a non-idealised configuration. The applicability of these ice-sheet models (in the sense of flowline+vertical) is extensively present in the literature. Let us now highlight a number of studies that exhibit the practical use of such a setup and provide insight on realistic ice sheets. Hindmarsh and Le Meur (2001) assessed the dynamical processes involved in the retreat of marine ice sheets, with a particular interest in West Antarctica after the Last Glacial Maximum. Haseloff and Segienko (2018) later considered the effect of buttressing on grounding line dynamics, thus corroborating the findings of existing numerical studies that the stability of confined marine ice sheets is influenced by ice-shelf properties. Other 2D ice-sheet models additionally employ real bedrock geometry sections. This is the case of Pattyn et al. (2006), who studied the role of transition zones in marine ice sheet dynamics and Jamieson et al. (2012), where ice stream stability was investigated on a reverse bed slope. The realism of the 2D setup can even account for glacial isostatic adjustment. To illustrate this, Payne (1995) studied limit cycles in the basal thermal regime of ice sheets considering a constant diffusivity of the asthenosphere. More recently, Bassis et al. (2017) investigated how Heinrich events are triggered by ocean forcing and modulated by isostatic adjustment. Even though ice shelves are not explicitly resolved in 2D models, the potential role of buttressing can be also considered via a parametrisation (Dupont and Alley, 2005; Schoof, 2007; Jamieson et al., 2012, Robel et al., 2014; 2018). We will elaborate on the introduction to emphasise the strong applicability of two dimensional ice-sheet models.

Information about the real world is not only obtained from spatially explicit models and simpler models allow the investigation of the relevance of specific processes. It is often

hard to discern the particular physical mechanism underlying certain behaviours for highly sophisticated 3D models, as the strong coupling among elements become an obstacle to isolate hypothesised causes. For this reason, 2D models are an extremely convenient tool to investigate the physical behaviour of ice sheets.

- *Applicability of the moving grid. As has been pointed out already in the original MIS-MIP publication, moving grids are very difficult to implement in two horizontal dimensions. This strongly limits the applicability of any findings produced with this model. Again, it might produce more accurate results for idealised-geometry benchmark experiments than other models do (at a similar resolution), but it cannot be used to e.g. find improved ways to represent the grounding line which could be applied in other models.*

Our findings show that active thermodynamics perturbs the hysteresis loop and the overall stability of an ice sheet. To do so, we extended MISMIP benchmarks to a thermally-active version where the system is forced via physical variables rather than an idealised ice rate factor. The particular grid over which the equations are discretised do not alter this behaviour, for that our results are not numerical artefacts of the chosen mesh. Therefore, we do not expect our results to change for a different grid discretisation.

More precisely, we expect the same physical behaviour as long as the ice viscosity varies upon temperature changes, irrespective of the chosen grid. The exact grounding line position may differ for a different grid, yet the physical mechanism underlying this mechanism remains unperturbed.

- *These two points together make me unsure what the added value of this model is over using other existing ice-sheet models that have a wider range of applications. E.g., if I hire a PhD candidate to do some work on the relation between thermodynamics and grounding-line migration, why would I advise them to use Nix instead of, say, CISM, PISM, Elmer, or any of the dozen other available models where such processes can be studied in both schematic and realistic cases?*

In all modelling areas there is an inevitable trade-off between a sophisticated physical description and fast computations. Generally, as the system is described with more realism, the necessary calculations demand strong computational resources. The main goal of this work is to present an ice-sheet model that overcomes this issue in certain cases.

There are several reasons for a researcher to use Nix model, among which we can highlight (summarised in Table 1):

1. **Thermomechanical coupling**. Nix explicitly resolves the energy balance by solving for the ice temperatures assuming a number of processes in the heat equation: vertical diffusion, horizontal and vertical advection, strain dissipation and basal frictional heat. As a result, the ice viscosity is thus temperature dependent. Moreover, the basal friction coefficient is also coupled to the thermal state of the base, thus accounting for a potential friction reduction if the pressure melting point is reached.

1. **Fast computations**. It combines a higher-order stress balance with active thermo-dynamics, while keeping extremely fast computations that allow for statistical studies with thousands of simulations involved, otherwise prohibited by 3D models.

   2a. For wall-clock times of the order of minutes, Nix allows for resolutions of $\Delta x = 0.1$ km (needed to properly resolve the grounding line) and simulated times of order $t \sim 10^3$ kyr. This extremely low computational cost allows for statistical studies with thousands of simulations involved, otherwise prohibited by 3D models.

   2b. Parallelisation. Nix users can optionally select parallel computing (supported by Eigen library), particularly convenient for high resolutions in the Blatter-Pattyn approximation, where large sparse matrices must be inverted. Moreover, it is possible to use Eigen's matrices, vectors, and arrays for fixed size within CUDA kernels.

   2c. It is hard to give a one-to-one comparison since other models that solve for the same higher-order momentum balance coupled with a thermodynamical solver use 3D solvers (partially providing Nix novelty). To give an estimation, the MALI ice-sheet model (Hoffman et al., 2018) control simulations averaged 5.26 simulated years per wall-clock hour. On the contrary, MISMIP experiments run with Nix reach $\sim 10^5$ simulated years per wall-clock hour on average. Thus, there is a 5-order magnitude difference in terms of computational time.

3. **Time stepping**. Unlike other models, Nix offers an adaptive time stepping based on the convergence of Picard's iteration in the velocity solution solver. This approach differs from the standard proportional-integral (PI) methods (e.g., Cheng et al., 2017) and strongly contributes to its fast computational performance in Nix.

4. **Friendly usage**. As a combination of low computational demands, a 2D setup and a clean structure, Nix can be even run on a regular laptop. Installation, compilation and execution are controlled from a simple Python program. This allows Nix to be used without deep knowledge of the C programming language (low-level, procedural and statically-typed). Even though Nix simulations can run on a personal computer, the user can exploit parallelization on a High Performance Computing cluster.

Table 1: Summary of Nix capabilities.

| Requirement | Nix capability |
| --- | --- |
| Fast computations | Low dimensionality, adaptive time-stepping, parallelization |
| Higher-order description | Blatter-Pattyn model |
| Thermomechanical coupling | Temperature-dependent ice viscosity and basal friction |
| Easy usage | Python wrapper |

In the revised manuscript, we have added some text to more specifically highlight these benefits and the potential use cases for Nix.

**Minor comments**

- *Line 2: "Nix is a 2D thermomechanical model...". It took me a few pages of reading to realise that you meant one horizontal plus one vertical dimension, rather than two horizontal dimensions. Please explain the "flowline+vertical" meaning of 2-D in the abstract.*

We agree that our description was somewhat confusing and will follow the suggestion.

- *Line 4: "...and shallow-ice." None of the experiments you present use the SIA. If the aim of the model is to study grounding-line dynamics, it seems unlikely that it will ever be used at all. If so, consider removing it from the description.*

Indeed, the SIA solver was merely included for the sake of completeness. We will note this more clearly in the revised text.

- *Line 6: "...including those of stochastic nature." This option is not used in any of your experiments. Either add relevant experiments or remove this statement.*

We will remove all mentions to stochastic boundary conditions capabilities as they are implemented but unused in the present study.

- *Line 6: "Nix has been verified for standard test problems" Please state here already that these are the MISMIP experiments.*

We will do so.

- *Line 15-16: "... Nix combines rapid computational capabilities..." You have not shown any results concerning computational capabilities. Either add relevant experiments or remove this statement.*

We will include a figure to show the computational speed as a function of the spatial resolution.

- *Line 21-23: I think the ISMIP intercomparison papers really need to be referenced here.*

We will do so.

- *Line 24: "... leading a number of authors to question their stability" Which authors?*

There is an extensive list of publications on this topic. Among others, we can highlight Bamber et al. (2009) Mouginot et al. (2014), Paolo et al. (2015), Feldmann and Levermann (2015), Shepherd et al. (2018), Rignot et al. (2019), Robel et al. (2019) and Pattyn and Morlighem (2020), Garbe et al. (2020), Jouhin et al.(2021) and Hill et al. (2023). We will include these references in the updated version.

- *Line 42: "... concluding that moving grid models are the most reliable..." Only two of the models in the first MISMIP paper used the flux condition ("Schoofing") approach, and if I recall, none of them used sub-grid friction scaling, both approaches that have since become commonplace in large-scale ice-sheet models (as opposed to moving grids, which I don't think any models use). Please discuss this.*

The reason why a moving grid is not employed in 3D models relies on the technical difficulties of implemention. Nevertheless, explicitly tracking the grounding line position and including it as part of the solution of the problem is still more convenient as shown in Pattyn et al. (2012), and, as in Nix, can be implemented in a flowline model without too much difficulty. A finite element discretisation with adaptive mesh refinement provides an alternative computation approach. We will expand our discussion on the implications of a flux condition ("Schoofing") and sub-grid friction scaling approaches.

- *Line 71-72: "... similar accuracy to the Blatter-Pattyn momentum equations..." While the leading error term might be second-order with respect to the aspect ratio epsilon*

*in both approximations, the velocity solutions produced by the DIVA are quite less accurate than those of the Blatter-Pattyn (try running the ISMIP-HOM experiments at the different length scales and you'll see a big difference when L < 20 km). Please include this nuance.*

Indeed, we will include this caveat.

- *Line 76-77: "... emerges as a clear outlier in terms of both model performance and its representation of the ice-flow physics itself." Robinson et al. (2022) did not include a Blatter-Pattyn solver in their comparison, so this statement is slightly misleading.*

We meant that DIVA outperforms all other solvers tested in the study. We will clarify this.

- *Line 81-82: "..., numerical simulations of these rapidly flowing bands are a well-known difficulty." Please elaborate on what difficulties with simulating ice streams are well known.*

This is stated in the first part of the sentence "*[...] the broad range of ice flow speeds observed in real ice sheets (Shepherd and Wingham, 2007; Truffer and Fahnestock, 2007; Vaughan and Arthern, 2007)*". This difficulty partially rests on the inability of ice-sheet models to simulate the observed large range of ice flow speeds. As noted by Bueler and Brown (2009), fast grounded ice flow is a combination of sliding over a hard/soft bed and shear deformation of the basal ice. Nevertheless, high-quality spatially distributed observation of near-base conditions are rare and constraining models becomes challenging. We will clarify this in the text.

- *Line 82: "Diverse approaches are found in the literature..." What approaches in what literature?*

Various modelling approaches have been considered to correctly model the large complexity in ice-stream dynamics. Tulaczyc et al. (2000) found that subglacial hydrology yields multiple modes of ice stream flow in a highly reduced model. Parameterizations of observed small scale phenomena (e.g., drainage networks) were later considered by coupling a flow band model and a simple hydrological model (Bougamont et al., 2003a; 2003b). Another flow band model was employed by van der Wel et al. (2013), additionally introducing a dynamic drainage model. We will elaborate on this briefly in the text.

- *Line 88-89: "It is a common approach to reduce the number of horizontal dimensions to the main flow direction so as to minimize computing time." This is only true for idealised-geometry experiments, not for realistic applications.*

There are several realistic applications published where models solve for two spatial dimensions (Hindmarsh and Le Meur, 2001; Haseloff and Sergienko, 2018), further employing real bedrock geometry sections (Pattyn et al., 2006; Jamieson et al., 2012) and glacial isostatic adjustment (e.g., Payne, 1995; Bassis et al., 2017). Additionally, the potential role of buttressing has been also considered in 2D models via a parametrisation (Dupont and Alley, 2005; Schoof, 2007; Jamieson et al., 2012; Robel et al., 2014; 2018).

Even so, we are fully aware of the limitations of a 2D setup. For this reason, the intended applications of the Nix model substantially differ from e.g., volume estimations. We will clarify the text to reflect this.

- *Line 91: "Unlike previous models..." The original MISMIP paper includes both a higher-order model and a full-Stokes model. Several existing 3-D ice-sheet models can use higher-order or full-Stokes dynamics with thermomechanical coupling (e.g. Elmer/ice, PISM, ISSM, UFEMISM). Please add some nuance here.*

There are indeed a number of models that use higher-order or full-Stokes dynamics with thermomechanical coupling. Nonetheless, they solve a far more computationally expensive 3D problem. Nix model demands far lower computational resources, offers an easy usage, straightforward visualisation and maintains the higher-order physical description of the system. We will include some nuance in the text.

- *Line 101-102: "...for efficiency and extremely fast computing..." Define "extremely fast".*

It is hard to give a one-to-one comparison since other models that solve for the higher-order momentum balance coupled with a thermomechanical solver are full 3D solvers (partially providing Nix novelty). To give an estimation, MALI ice-sheet model (Hoffman et al., 2018) control simulations averaged 5.26 simulated years per wall-clock hour. On the contrary, MISMIP experiments run with Nix reach $\sim 10^5$ simulated years per wall-clock hour on average. Thus, there is a 5-order magnitude difference in terms of computational time.

- *Line 103: "...NetCDF and Eigen libraries" Please include references for these libraries. Also, do any of them make use of parallel computing?*

Nix users can optionally select parallel computing (supported by Eigen library), particularly convenient for high resolutions in the Blatter-Pattyn approximation, where large sparse matrices must be inverted. Moreover, it is possible to use Eigen's matrices, vectors, and arrays for fixed size within CUDA kernels. We will include references to NetCDF and Eigen libraries.

- *Line 113: "Our system is thought to thermodynamically evolve in time..." A strange way to phrase this.*

We will rephrase.

- *Line 128: "...extremely high spatial resolutions..." Define "extremely high".*

We refer to $\Delta x \sim 0.5$ km. We will be explicit in the manuscript.

- *Eq. 11: The basal drag coefficient beta does not appear in any other equations. Is beta an input use to calculate $c_b$ in Eqs. 9 and 10?*

No, $\beta$ is in fact calculated in Nix and $c_b$ is an input coefficient. We wanted to follow the standard notation for basal friction as $\tau = \beta u$, where $\beta(u)$ can be a function of the velocity. We will clarify this in the text.

- *Eq. 18: the squiggly rho (ratio of densities) is difficult for me to distinguish from the regular rho (density). As you simply write out $\rho_w/\rho$ in e.g. Eqs. 19, 21 and 22, consider doing so here as well and removing squiggly rho altogether.*

The symbol $\varrho = \rho/\rho_w$ was solely introduced to lighten the expressions. We will be consistent with the notation and update the manuscript accordingly.

- *Eq. 18: what does S stand for?*

S stands for surface mass balance (already noted in line 171).

- *Eqs. 21-22: the melt rate M does not appear anywhere in the continuity equation. Also, T0 in these equations in my understanding indicates the pressure+salinity-corrected (and therefore depth-dependent) ocean freezing temperature, is this also depth-dependent in your model?*

Following Christian et al. (2022), we include the melt rate $M$ in the ice flux computation as an additional term at the terminus position.

Regarding $T_0$, the model currently works with temperature anomalies $\Delta T$ that determine the melt rate $M$. Since this term is included via ice flux calculations that consider the vertically-integrated velocity (i.e., $q = \bar{u}H$), there is no depth-dependency of temperature anomalies at the present moment. We will include a statement to clarify this.

- *Line 263: "Oceanic melting beneath ice shelves..." Earlier, you explained that in the flowline case (which assumes no buttressing), the geometry of the shelf does not affect the flow of the grounded ice, and that therefore you do not need to model a shelf. How then can sub-shelf melt affect your model? Is this included as a negative mass balance term at the last grid point of your grounded domain, as a sort of frontal melt rate? If so, how is this derived from the sub-shelf melt rate?*

The sub-shelf melt is included as an additional term in the ice flux computation (see our previous answer). The ice thickness then adjusts to this increased outflow of ice at the grounding line via the mass continuity equation (see also supplementary material in Christian et al., 2022).

- *Line 280-281: "...the grid points distribution yields higher resolution near the grounding line following a polynomial or an exponential law." Please provide this law.*

The law will be provided in the Appendix.

- *Line 283: "...moving grid models are presumably the best choice from a numerical perspective..." Please clarify that this does not apply to models with two horizontal dimensions.*

We will include a statement to clarify this. To the authors' knowledge, a moving grid mesh has been only applied to one horizontal dimension, upon which it outperforms the other discretisation choices as shown in Pattyn et al. (2012).

- *Figure 2: It is unclear on which grid points you define velocities, and on which you define ice thicknesses/temperatures. Also, the caption states that "..., the position of the last horizontal point (r - 1/2) explicitly tracks the grounding line L(t)". Does that mean that grid point r is floating?*

No, the point $r$ does not exist as we start counting in 0 (Fig. 2). The point $r - 1/2$ is the very last point and it explicitly tracks the position of the grounding line. We follow an Arakawa grid type C. Velocities are thus evaluated at the centers of grid faces and scalar magnitudes (e.g., ice thickness and temperatures) are computed at the grid centres.

- *Section 6: at what horizontal and vertical resolutions do you run your simulations? What kind of error would you expect based on the convergence tests you mention (but do not show) later on?*

Simulations are run at $\Delta x = 2$ km. We have found negligible differences among simulations with $\Delta x \le 5$ km.

- *Figure 3b: does this include both the advancing and retreating phases of the experiment?*

Yes, both branches coincide and points overlap. We will clarify this in the text.

- *Line 405: "... near the melting point (Fig. 4d)." This should be Fig 5d.*

This is a typo indeed. Thank you.

- *Line 408: "... minimum temperature of $23^o C$ (Fig. 4c)." This should be Fig. 5c.*

This typo will be fixed. We thank the reviewer.

- *Figure 5: panels A and B are not referenced in the text.*

References will be included in the manuscript.

- *Line 424: "Figure 7 illustrates the high sensitivity to that stems from the heat exchange velocity parameter gamma." I find it difficult to understand the goal of this experiment. If I understand correctly that you use the melt rate M as a sort of frontal melt/calving rate, then M is a scalar number which scales linearly with gamma. So if you were to put M on the horizontal axis, then the curves of the four experiments should overlap, correct?*

The goal of this experiment is to show the strong sensitivity to the heat exchange velocity parameter $\gamma$ range of values given in Favier et al. (2019). When presented as a function of $M$, the four experiments should overlap. Nonetheless, the main point here is the large range in temperature anomalies at which the ice-sheet collapse occurs depending on the particular $\gamma$ value.

- *Figure 6b: how much time is there between the steps in ocean temperature? It looks like about 30,000 years, is that enough for the temperature to reach a steady state? In my experience the ice geometry itself equilibrates quite a lot faster. What would you expect to see if you reduce the time between the steps, so you deliberately prevent the temperature from reaching equilibrium? This is where the added value of thermomechanical coupling would really appear.*

Yes, precisely 30 kyr. We will state this clearly in the text. This step length allows for the ice ice geometry to reach a steady-state as these experiments are intended to be quasi-equilibrium simulations (as MISMIP). If we were to reduce the time between steps, the hysteresis loop would then be a transitory response given that the ice temperature may not adjust to the new geometry. The added value of the thermomechanical coupling does not solely rely on the transitory response (i.e., thermal inertia), but also on the perturbed stability of the quasi-equilibrium hysteresis loop. In other words, the thermomechanical coupling already determines the stable regions in a quasi-steady description and not only through the effects of thermal inertia. Our goal here is to first show this more fundamental behaviour. But, of course, a further benefit will be to be able to study transient responses too, which we will highlight in the discussion.

- *Line 439: "... a sensitivity test to spatial resolution (not shown)..." Why do you not show this? I'm actually quite interested. The stress-free boundary condition to the Blatter-Pattyn approximation at the ice surface is very tricky to implement, and I'm curious to see what order of convergence you get.*

Convergence of a number of Stokes approximations to the semi-analytical results of Schoof (2007) was already presented in MISMIP experiments (Pattyn et al., 2012). However, we will include an additional figure showing the convergence towards the analytical solution.

- *Line 459: "Setting the vertical advection at the surface equal to the accumulation (0.3 m/yr) is a standard choice..." What do you mean by this? It is my understanding that most large-scale ice-sheet models these days explicitly solve the heat equation in three dimensions, with the vertical velocity that appears in the advection term resulting from vertically integrating the 3-D horizontal ice velocity field (i.e. conservation of mass for incompressible ice). This is not so much a choice as it is simply a direct consequence of basic physics.*

Vertical velocity can be indeed calculated from the incompressibility of the Stokes flow. We meant that the analytical solutions presented in Moreno-Parada et al., (2022) consider a prescribed vertical velocity profile where the surface value equals the accumulation rate (0.3 m/yr). This paragraph will be rephrased for clarity.

- *Line 473: "... stochastic boundary conditions capability..." You have not shown anything relating to his.*

We will remove all mentions to stochastic boundary conditions capabilities as they are implemented but unused in the present study.

- *Line 491-493: "More generally ... and viscosity" What does this imply for models of realistic ice sheets? It is still not uncommon for models in the ISMIP6 ensemble to neglect evolving ice temperatures in future projections of the Antarctic ice sheet. Do your findings imply that this introduces a significant error/bias in the results of these models?*

ISMIP6 projections reached the year 2100. From a purely thermodynamic point of view of the Antarctic Ice Sheet, $\sim 100$ years is too short a time span to yield notable changes in the thermal structure of the ice sheet. We must stress that experiments carried out in this work range a simulated time of $\sim 10^2$ kyr. Therefore, we would not expect significant errors in such a short timescale.

- *Appendix A: there are two Appendix A2's, please fix this.*

This typo will be fixed.

- *Line 510: "The position in the spatial coordinates is then given by..." These expressions are not correct when the grid is irregular (and $\Delta\sigma_i$ is indeed a function of i). Also, time is not a spatial coordinate.*

Thank you for pointing out this mistake. The position is instead give by: $\sigma_i = \sum_{k=0}^{i} \Delta\sigma_k$ and $\zeta_i = \sum_{k=0}^{i} \Delta\zeta_k$. The Appendix will be updated accordingly.

- *Line 522: "We thus have a linear system of $6 \times r \times p$ unknowns..." Don't you mean r*p unknowns, interrelated by a matrix with 6*r*p non-zero coefficients?*

Indeed, we will fix this typo.

- *Appendix A1: Please also provide the discretisation scheme you used for the boundary conditions.*

Nix uses a standard two-point discretisation for the top boundary condition. We will expand the appendix to include it.

- *Appendix A5: adaptative = adaptive*

This typo will be fixed.

- *Appendix B: Either include some experiments with stochastic forcing, or remove this text.*

We will remove all mentions to stochastic boundary conditions capabilities as they are implemented but unused in the present study.

---

## Referee Report (RR1)

**2^(nd) review of Moreno-Parada et al.**

**By Tijn Berends**

After the first round of reviews, the authors have added references to earlier studies involving flowline models, in order to demonstrate the practical applicability of the Nix model. They have added extra information about their discretisation and included some extra text discussing the relevance of their results in the thermomechanically coupled experiments. This has generally improved the quality of the manuscript. However, I still have a few concerns which I believe should be addressed before publication.

**Major comments**

**1. Applicability**

I think you could still do a better job of convincing the reader of the practical value of the Nix model. The experiments presented in the manuscript used "a horizontal resolution of $\Delta x$ = 2 km and 35 vertical layers". These numbers would not pose any difficulty in terms of performance for existing 3-D models (PISM, CISM, Elmer, UFEMISM, etc.) when using a flow-band set-up, so the added value of Nix in this application does not really stand out. However, in your rebuttal, you state that "For wall-clock times of the order of minutes, Nix allows for resolutions of $\Delta x$ = 0.1 km ... and simulated times of order t ~ 10^3 kyr.". This resolution is 20 times higher than what is used in your experiments. Such a high resolution likely cannot be easily matched by existing 3-D models (at least, not without using really large numbers of CPU cores), and so there the added value of Nix would be much more noticeable. I think that providing some numbers to demonstrate Nix's performance at such high resolutions, including some (hypothetical) examples of experiments where such a high resolution would be necessary, would help convince the reader of why (as I asked in my first review), they should use Nix instead of any of the already existing ice-sheet models.

**2. Discrepancies in new results**

The results of the thermomechanically coupled experiments in the current version of the manuscript are quite different from those in the previous version. It is not entirely clear to me exactly what has changed that could cause these differences, nor why these changes were made. There are also a few discrepancies in the new results which need to be explained.

In Sect. 6.2.,1 I see that you added an altitude dependency to the surface temperature forcing, why was this done? Also, in the authors' response, you mention "both adiabatic and moist lapse rates", but only one value of 9.8 K/km is given in the manuscript. Also, in the previous version, you needed to reduce the surface temperature by about 80 K to cause the ice sheet to advance, whereas now you only need a 30 K change. Can you explain this difference?

In the new version of Fig. 5, large parts of the ice in panel E (the lower left, near the bedrock and near the ice divide) seem to be about 20 to 30 K colder than in panel A. In the text, you

claim that the 30 kyr steps are long enough to "ensure thermal quasi-equilibrium", and since the geometries in these two panels are identical, the temperature profiles should be identical too. Can you explain where these substantial temperature differences come from?

**Minor comments**

L76-77 "...though differences are particularly notable for resolutions below 20 km." The differences between the DIVA and the BPA arise when the scale of subglacial bedrock topography is smaller than ~20 km (in the ISMIP-HOM experiments). This has nothing to do with resolution, nor with any other numerical model parameter. It is a consequence of the neglected strain rates in the DIVA, making the velocity errors increase faster with the aspect ratio of the ice than they do in the BPA.

L87 "...ice high-quality spatially distributed observations..." unclear what you mean by this.

L307 "This melt rate is included as an additional term in the ice flux computation (Eq. 7 and 18)" Please provide the modified versions of these two equations including M.

L397 "...(Fig. 3c and 3c)..." I think these are 3c and 3d.

L439 "...panels 5a, 5b 5e..." I think these are 5a, 5c, and 5e.

L550 "...the required calving at the grounding line..." I thought M was supposed to represent some sort of melt?

---

## Author Response (AR2)

**Authors' response**
**EGUSPHERE-2023-2690**

Daniel Moreno-Parada, Alexander Robinson,
Marisa Montoya, and Jorge Alvarez-Solas.

December 16, 2024

**Contents**

Note: Reviewers' comments are given in italic font whereas the authors' responses read in regular font.

**1 Editor's remarks**

The authors are thankful to the editor (Ludovic Räss) for his comments, thus allowing an improved version of the paper. We have addressed the main points raised by the editor:

- Better define, motivate and showcase the applicability of Nix's new features when compared to existing models.

- Explain potential discrepancies with the previous version.

- Updated GitHub repository link: `https://github.com/d-morenop/nix`.

- Removed typos.

The result of this revision is evident in a clearer and more convincing version of the paper. Nix thus presents itself as an extremely versatile model combining usage simplicity, low computational costs and high-order physics at extremely high resolution ($\Delta x < 0.1$ km).

**2 Tijn Berends**

The authors are grateful to the reviewer for their constructive comments. We now provide our answers (regular font) to the main concerns raised by the reviewer (italic).

**Major comments**

- *Applicability. I think you could still do a better job of convincing the reader of the practical value of the Nix model. The experiments presented in the manuscript used "a horizontal resolution of $\Delta x = 2$ km and 35 vertical layers". These numbers would not pose any difficulty in terms of performance for existing 3-D models (PISM, CISM, Elmer, UFEMISM, etc.) when using a flow-band set-up, so the added value of Nix in this application does not really stand out. However, in your rebuttal, you state that "For wall-clock times of the order of minutes, Nix allows for resolutions of $\Delta x = 0.1$ km ... and simulated times of order $t \sim 10^3$ kyr.". This resolution is 20 times higher than what is used in your experiments. Such a high resolution likely cannot be easily matched by existing 3-D models (at least, not without using reallylarge numbers of CPU cores), and so there the added value of Nix would be much more noticeable. I think that providing some numbers to demonstrate Nix's performance at such high resolutions, including some (hypothetical) examples of experiments where such a high resolution would be necessary, would help convince the reader of why (as I asked in my first review), they should use Nix instead of any of the already existing ice-sheet models.*

We thank the reviewer for pointing out Nix high computational speed. Appendix H has been updated by including a figure that shows the computational speed time at high resolutions, i.e., up to $\Delta x = 0.05$ km, far beyond existing 3D models (see Fig. 1 in this document and Appendix H of the updated manuscript).

[Figure]

Figure 1: Nix computational speed for the three solvers available. The double $x$-axis represents the number of horizontal grid points $n$ and the corresponding resolution at the grounding line $\Delta x$. Note that Nix allows for an unevenly-spaced stretched grid that explicitly tracks the grounding line position. This figure is now part of Appendix H.

Furthermore, the practical use of Nix ice-sheet model does not solely lie on its high-resolution performance, but also in the gap filled in the model hierarchy spectrum. Nix is a 2D marine-terminating ice-sheet model described by the higher-order Blatter-Pattyn stress balance and fully thermodynamically coupled. Unlike previous existing 2D models, Nix solves for the ice temperatures and evaluates the importance of ice thermomechanics for stability and grounding line migration by forcing with physical variables: air temperatures and oceanic anomalies. To the authors' knowledge, this represents a novel result and sheds light on the stability of marine-terminating ice sheets. Previous studies, such as those focused on attribution exercises to anthropogenic-induced ice-sheet retreat (e.g., Christian et al., 2022), use simplified physics that neglect the thermochemical coupling and are consequently biased by unrealistic constant temperatures both in space and time. Other examples of neglected thermal effect on ice viscosity also involve Heinrich Events triggered by oceanic warming (Bassis et al., 2017).

Nix usage stands out for its simplicity. Highly complex 3D ice-sheet models (e.g., ElmerIce, ISSM) require large efforts for installation, computational resources and careful preparation of input fields. On the contrary, only two command lines are enough to run Nix:

```
1 $ git clone https://github.com/d-morenop/nix
2 $ python run_nix.py
```

This will run the desired experiment (within a few minutes) solving the higher-order Blatter-Pattyn stress balance. Then, simply by running the script nix_plot.py, the user can visualize the output saved in the NetCDF file "*nix.nc*". Alternatively, one can employ the ncview tool for a direct inspection of the simulation output.

- *Discrepancies with new results.*

This point was already noted by the authors in the previous response (see the uploaded document "Relevant changes made in the manuscript"; 10th of May, 2024). We quote here what was already written for clarification: "*[...] the authors further noted a misconception on the previous diagnosis of the thermomechanically coupled simulations [...]. Results are now correctly interpreted and both Section 6.2.2 and Fig. 6 have been updated accordingly showing that the hysteresis loop in overdeepening beds is in fact widen when thermodynamics is considered*".

In the old manuscript version, results corresponded to simulations in which ice temperatures were calculated, but the viscosity was not updated accordingly. This led to a different hysteresis loop. All figures and results were updated as it was already noted in the last review. This explains all figures discrepancies between the last two versions.

- *In Sect. 6.2.,1 I see that you added an altitude dependency to the surface temperature forcing, why was this done? Also, in the authors' response, you mention "both adiabatic and moist lapse rates", but only one value of 9.8 K/km is given in the manuscript. Also, in the previous version, you needed to reduce the surface temperature by about 80 K to cause the ice sheet to advance, whereas now you only need a 30 K change. Can you explain this difference?*

In the first manuscript version, the applicability and realism of the modeled were criticized. To overcome this issue, an adiabatic lapse rate was simply included to further improve realism and applicability. The corresponding value of the moist lapse rate has been included in the manuscript.

The difference in temperature amplitude to produce advance in the ice sheet shares the same answer that we elaborated above: old results corresponded to simulations in which ice temperatures were calculated, but the viscosity was not updated accordingly. Now that the ice viscosity is also updated with the new temperatures, the ice velocity adjusts accordingly and the amplitude in the atmospheric forcing necessary to produce advance is thus reduced.

- *In the new version of Fig. 5, large parts of the ice in panel E (the lower left, near the bedrock and near the ice divide) seem to be about 20 to 30 K colder than in panel A. In the text, you claim that the 30 kyr steps are long enough to "ensure thermal quasi-equilibrium", and since the geometries in these two panels are identical, the temperature profiles should be identical too. Can you explain where these substantial temperature differences come from?*

Thank you for the comment. Fig. 5e did not correspond to the very last frame for the forcing. As the reviewer notes, the temperatures profiles are expected to be identical given the boundary conditions of the problem. We have updated accordingly the figure.

**Minor comments**

- *L76-77 "...though differences are particularly notable for resolutions below 20 km." The differences between the DIVA and the BPA arise when the scale of subglacial bedrock topography is smaller than 20 km (in the ISMIP-HOM experiments). This has nothing to do with resolution, nor with any other numerical model parameter. It is a consequence of the neglected strain rates in the DIVA, making the velocity errors increase faster with the aspect ratio of the ice than they do in the BP.*

The text does not discuss the physical reason why these differences appear. It is simply stated that such deviation only becomes noticeable for resolutions higher than 20 km. As the reviewer later elaborates: "[...] the velocity errors increase faster with the aspect ratio of the ice than they do in the BP". Consequently, DIVA and Blatter-Pattyn stress balances are virtually equivalent for resolutions coarser than 20 km within the ISMIP-HOM experimental setup.

- *L87 "...ice high-quality spatially distributed observations..." unclear what you mean by this.*

Remote sensing has important limitations in terms of spatially distributed observations of basal conditions. This poses a restriction on how we constrain ice-sheet models in fast flowing regions. We refer the reviewer to Bueler and Brown (2009) for further details. The manuscript has been updated for clarification.

- *L307 "This melt rate is included as an additional term in the ice flux computation (Eq. 7 and 18)" Please provide the modified versions of these two equations including M.*

We have updated the Discretization scheme (Eq. D3) to account for this potential additional term.

- *L397 "...(Fig. 3c and 3c)..." I think these are 3c and 3d.*

Indeed. The typo has been fixed.

- *L439 "...panels 5a, 5b 5e..." I think these are 5a, 5c, and 5e.*

Indeed. The typo has been fixed.

- *L550 "...the required calving at the grounding line..." I thought M was supposed to represent some sort of melt?*

Following Christian et al., (2022), we simply interpret this term as flux anomalies at the terminus driven by variable ocean conditions. In real glaciers, these anomalies could be driven by variable calving, submarine melt, or a combination. This is now explicitly stated in the manuscript (Lines 307-309).

**References**

Bueler, E., and J. Brown (2009), Shallow shelf approximation as a "sliding law" in a thermomechanically coupled ice sheet model, J. Geophys. Res., 114, F03008, doi:10.1029/2008JF001179

Christian, J. E., Robel, A. A., and Catania, G.: A probabilistic framework for quantifying the role of anthropogenic climate change in marine-terminating glacier retreats, The Cryosphere, 16, 2725–2743, https://doi.org/10.5194/tc-16-2725-2022, 2022.

Pattyn, F., Perichon, L., Aschwanden, A., Breuer, B., de Smedt, B., Gagliardini, O., Gudmundsson, G. H., Hindmarsh, R. C. A., Hubbard, A., Johnson, J. V., Kleiner, T., Konovalov, Y., Martin, C., Payne, A. J., Pollard, D., Price, S., Rückamp, M., Saito, F., Souček, O., Sugiyama, S., and Zwinger, T.: Benchmark experiments for higher-order and full-Stokes ice sheet models (ISMIP–HOM), The Cryosphere, 2, 95–108, https://doi.org/10.5194/tc-2-95-2008, 2008.

Bassis, J., Petersen, S. and Mac Cathles, L. Heinrich events triggered by ocean forcing and modulated by isostatic adjustment. Nature 542, 332–334 (2017). https://doi.org/10.1038/nature21069

Payne, A. J. (1995), Limit cycles in the basal thermal regime of ice sheets, J. Geophys. Res., 100(B3), 4249–4263, doi:10.1029/94JB02778.

---

## Author Response (AR3)

**Authors' response**
**EGUSPHERE-2023-2690**

Daniel Moreno-Parada, Alexander Robinson,
Marisa Montoya, and Jorge Alvarez-Solas.

February 27, 2025

**Contents**

**1 Editor's remarks**

The authors are thankful to the editor (Ludovic Räss) for his comments, thus allowing an improved version of the paper. We have addressed all points raised by the editor:

- Highlight where the high-resolution is mandatory to converge some model configurations (showing, e.g., that physical quantity of interest converges to a steady value upon reducing grid size)

  - A new Appendix B has been described where Nix model convergence is detailed, reaching resolutions of 60 metres. Fig. 1 in this document (Fig. B1 in the manuscript) depicts a number of physical variables are plotted as a function of grid size. Results are then compared to the analytical solution presented by Schoof (2007).

- Further development on the pros and cons of the (parallel) implementation.

  - We have entirely written a new section (Section 7: Model scalability and performance) to report the computational speed (Fig. 8 in the manuscript) and the details regarding the parallel implementation.

- Additional performance experiments such as, e.g., effect of using multiple core via OpenMP, weak and/or strong scaling.

  - In Section 7, there are two new figures (Figs. 2 and 3 in this document, and Figs. 9 and 10 in the revised manuscript) where both strong and weak scalability of Nix are tested, respectively. Several parameter permutations of the linear solver are considered: the total number of iterations $N$ and the optimization level level during compilation with OpenMP flag (i.e., -O1, -O2 and -O3).

- Updated GitHub and Zenodo repositories. Link: `https://github.com/d-morenop/nix`.

- Removed typos.

[Figure]

Figure 1: Convergence study with Nix model. From top to bottom: ice velocity, terminus position, ice thickness and ice flux. All variables are evaluated at the grounding line. The double $x$-axis denotes the total number of horizontal grid points $n$ and the corresponding spatial resolution $\Delta x$ given the stretched coordinate transformation.

[Figure]

Figure 2: Acceleration and efficiency for strong scalability experiments. The maximum number of iterations $N$ in the sparse linear problem is given as a colour legend. Line styles denote the three levels of optimization provided by OpenMP during compilation (O1, O2 and O3, in increasing order).

[Figure]

Figure 3: Efficiency for weak scalability experiments. The maximum number of iterations $N$ in the sparse linear problem is given as a colour legend, ranging from 10 to $10^6$. Line styles denote the three levels of optimization provided by OpenMP during compilation (O1, O2 and O3, in increasing order).

**References**

Schoof, C. (2007), Ice sheet grounding line dynamics: Steady states, stability, and hysteresis, J. Geophys. Res., 112, F03S28, `https://doi.org/10.1029/2006JF000664`, 2007.

Gagliardini, O., Zwinger, T., Gillet-Chaulet, F., Durand, G., Favier, L., de Fleurian, B., Greve, R., Malinen, M., Martín, C., Råback, P., Ruokolainen, J., Sacchettini, M., Schäfer, M., Seddik, H., and Thies, J.: Capabilities and performance of Elmer/Ice, a new-generation ice sheet model, Geosci. Model Dev., 6, 1299–1318, `https://doi.org/10.5194/gmd-6-1299-2013`, 2013.

Pattyn, F., Perichon, L., Aschwanden, A., Breuer, B., de Smedt, B., Gagliardini, O., Gudmundsson, G. H., Hindmarsh, R. C. A., Hubbard, A., Johnson, J. V., Kleiner, T., Konovalov, Y., Martin, C., Payne, A. J., Pollard, D., Price, S., Rückamp, M., Saito, F., Souček, O., Sugiyama, S., and Zwinger, T.: Benchmark experiments for higher-order and full-Stokes ice sheet models (ISMIP–HOM), The Cryosphere, 2, 95–108, `https://doi.org/10.5194/tc-2-95-2008`, 2008.

Fischler, Y., Rückamp, M., Bischof, C., Aizinger, V., Morlighem, M., and Humbert, A.: A scalability study of the Ice-sheet and Sea-level System Model (ISSM, version 4.18), Geosci. Model Dev., 15, 3753–3771, `https://doi.org/10.5194/gmd-15-3753-2022`, 2022.

---

## Author Response (AR4)

**Authors' response**
**EGUSPHERE-2023-2690**

Daniel Moreno-Parada, Alexander Robinson,
Marisa Montoya, and Jorge Alvarez-Solas.

March 18, 2025

**Contents**

**1 Editor's remarks**

The authors are thankful to the editor (Ludovic Räss) for his comments, thus resulting on a more detailed discussion section (Section 7 in the manuscript). We herein clarify some technical aspects raised by the editor. Editor's remarks are given in italic, whereas the authors' answers read in regular font.

- *Regarding the technical aspects, could you clarify the versioning of the software?*

Thank you for the suggestion. Nix GitHub repository did not have any tags whatsoever. The authors have now tagged the current version as 1.0, where potential future changes will follow SemVer. We have updated the repository as well as the default main branch when cloning into Nix.

- *Moreover, you may want to enhance slightly your README(s) as to better convey the information and important steps the users need to follow in order to get started. Being in Markdown format, please add code snippets and/or blocks where needed, and make use of (sub-)section separators to enhance readability - Thanks!*

The README file has been updated accordingly. Readability was also enhanced exploiting the Markdown format. Nix GitHub repository also reflects these last changes.

- *It would be interesting to provide some directions or hint on why there is only a marginal performance increase when using threads > 1. As you explain, the heavy lifting is done using the Eigen library. May it be that this library calls e.g. into BLAS under the hood and utilises already more than one thread by default? Any addition on this topic would be beneficial to the reader.*

There are a wide range of reasons that may explain a marginal performance when parallelization is considered. Before diving into the details, we must note that Eigen library does not call BLAS. Instead, Eigen's native multithreading uses OpenMP (de thafault) or Intel TBB.

Given that the velocity solution carries the heaviest computations and it relies on the BiCGSTAB method. Moreover, we have additionally tested the scalability performance on a slightly simpler approach: the Conjugate Gradient method (CG). Results are consistent for both algorithms and suggest that the bottleneck is given by the limitations of sparse matrix-vector multiplications. We now discuss potential limitations concerning the particular linear solvers employed in Nix:

- First, it is important to note that the CG and BiCGSTAB approaches are memory-bound, and not compute-bound. As a result, performance is limited by memory access speed rather than computational power. Unlike dense matrix-matrix multiplication, CG and BiCGSTAB involve two main types of computations:

    - Sparse matrix-vector products (SpMV).
    - Vector updates (i.e., dot products and norms).

    These operations do not parallelize as effectively since they involve multiple memory accesses and low arithmetic intensity.

- Difficulties with parallel sparse Matrix-Vector Multiplication (SpMV). In both CG and BiCGSTAB, the dominant operation is SpMV and it is hard to parallelize for a number of reasons:

    - It involves random memory access patterns that do not benefit from CPU cache locality (recall that sparse matrices are not stored in continuous block of memory). As a result, each CPU frequently loads new, non-contiguous data into the cache.
    - It is bandwidth-limited: CPUs are often limited by how fast they can fetch data from RAM rather than by raw FLOPS.
    - Load balancing issues: different rows of a sparse matrix have different numbers of nonzeros entries, leading to workload imbalance among threads.

    Eigen method remains simple and centred on a data-parallel approach (i.e., partitioning the sparse matrix). Namely, rows are divided among the available threads leveraged by OpenMP and then, each thread processes a contiguous block of rows to minimize the synchronization overhead. Since the number of non-zero entries per row remains constant, the partitioning approach seemed justified. Nevertheless, Eigen does not allow for more complex partitioning strategies such as graph-based approaches (Çatalyürek and Aykanat, 1999), corner partitioning (Wolf et al., 2008) or block-cyclic partitioning (Aboelaze et al., 1991; Petitet and Dongarra, 1999) to minimize the total communication volume while keeping the computation balanced across compute nodes.

- Potential overhead. Since the BiCGSTAB method computes two matrix-vector operations per iteration, there is an increased communication overhead in distributed settings. For small to medium-sized problems, the overhead of creating/managing threads may negate some potential speedup. This is of particular relevance for the problem at hand: a 2D ice-sheet model. The Blatter-Pattyn stress balance implies a two-dimensional velocity field, considerably less computationally expensive than the three-dimensional counterpart. On the other hand, CG only computes one matrix-vector multiplication (Shewchuk, 1994), thus decreasing the communication. Even so, both methods show a similar efficiency when parallelization is enabled, suggesting that overhead is not the main cause of marginal speed-up.

Overall, the potential causes for a marginal speed-up in parallel runs mainly concern the solver employed in the linear problem. Both the size of the physical problem and the extremely well optimized memory allocation (as shown in serial runs) yield a rapid decrease in efficiency. For completeness, a similar scalability analysis was also performed for the CG algorithm and results are consistent. The maximum speed-up remains marginal, reaching values of $\sim 2.5$. This further supports the hypothesis that SpMV is the limiting component when enabling parallelization. We suggest that Eigen simplicity of a row-wise partitioning of the sparse matrix is not enough to yield optimal parallelization. Future work is needed to explore more advanced approaches of the partitioning strategy.

As a conclusion, the discussion herein presented emphasizes that Nix ice sheet model is designed to study a complex system with minimum resources: within a few hours, a 100-metres-resolution Blatter-Pattyn stress balance coupled with full thermodynamics is feasible even on a regular laptop. The discussion section of the manuscript version has been updated to reflect the causes explored by the authors.

**References**

Petitet, A., and Dongarra, J.J. (1999). Algorithmic Redistribution Methods for Block-Cyclic Decompositions. IEEE Trans. Parallel Distributed Syst., 10, 1201-1216.

M. Aboelaze, N. Chriso choides, and E. Houstis. The Parallelization of Level 2 and 3 BLAS Op erations on Distributed Memory Machines. Technical Rep ort CSD-TR-91-007, Purdue University, West Lafayette, IN, 1991.

U. Çatalyürek and C. Aykanat. Hypergraph-partitioning-based decomposition for parallel sparse-matrix vector multiplication. IEEE Trans. Parallel Dist. Systems, 10(7):673–693, 1999.

Shewchuk, J. R. (1994). An Introduction to the Conjugate Gradient Method Without the Agonizing Pain. Technical Report. Carnegie Mellon University, USA.

Schoof, C. (2007), Ice sheet grounding line dynamics: Steady states, stability, and hysteresis, J. Geophys. Res., 112, F03S28, `https://doi.org/10.1029/2006JF000664`, 2007.

Fischler, Y., Rückamp, M., Bischof, C., Aizinger, V., Morlighem, M., and Humbert, A.: A scalability study of the Ice-sheet and Sea-level System Model (ISSM, version 4.18), Geosci. Model Dev., 15, 3753–3771, `https://doi.org/10.5194/gmd-15-3753-2022`, 2022.

Wolf, Michael M., Bruce Hendrickson and Erik G. Boman.: Optimizing parallel sparse matrix-vector multiplication by partitioning, 2008.